# ManiSkill2: A Unified Benchmark for Generalizable Manipulation Skills

**Jiayuan Gu[1†], Fanbo Xiang[1†], Xuanlin Li[1*], Zhan Ling[1*], Xiqiang Liu[1*], Tongzhou Mu[1*] Yihe Tang[1*], Stone Tao[1*], Xinyue Wei[1*], Yunchao Yao[1*], Xiaodi Yuan[2], Pengwei Xie[2] Zhiao Huang[1], Rui Chen[2], Hao Su[1]**
[1]University of California San Diego, [2]Tsinghua University

## Abstract

Generalizable manipulation skills, which can be composed to tackle long-horizon and complex daily chores, are one of the cornerstones of Embodied AI. However, existing benchmarks, mostly composed of a suite of simulatable environments, are insufficient to push cutting-edge research works because they lack object-level topological and geometric variations, are not based on fully dynamic simulation, or are short of native support for multiple types of manipulation tasks. To this end, we present ManiSkill2, the next generation of the SAPIEN ManiSkill benchmark, to address critical pain points often encountered by researchers when using benchmarks for generalizable manipulation skills. ManiSkill2 includes 20 manipulation task families with 2000+ object models and 4M+ demonstration frames, which cover stationary/mobile-base, single/dual-arm, and rigid/soft-body manipulation tasks with 2D/3D-input data simulated by fully dynamic engines. It defines a unified interface and evaluation protocol to support a wide range of algorithms (e.g., classic sense-plan-act, RL, IL), visual observations (point cloud, RGBD), and controllers (e.g., action type and parameterization). Moreover, it empowers fast visual input learning algorithms so that a CNN-based policy can collect samples at about 2000 FPS with 1 GPU and 16 processes on a regular workstation. It implements a render server infrastructure to allow sharing rendering resources across all environments, thereby significantly reducing memory usage. We open-source all codes of our benchmark (simulator, environments, and baselines) and host an online challenge open to interdisciplinary researchers.

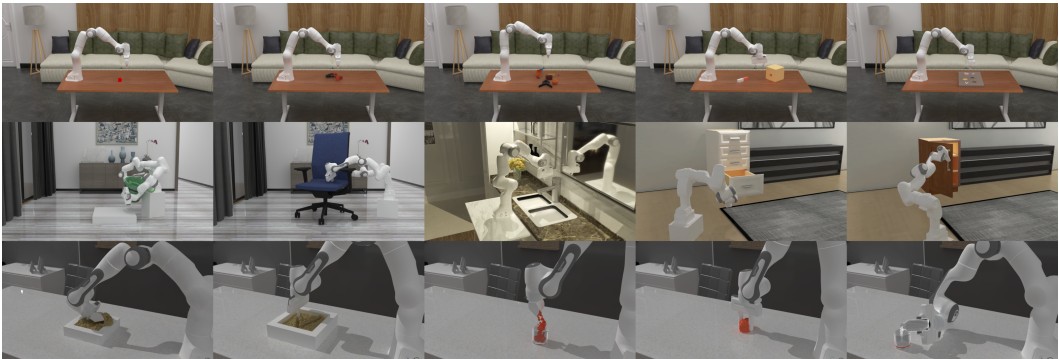

Figure 1: ManiSkill2 provides a unified, fast, and accessible system that encompasses well-curated manipulation tasks (e.g., stationary/mobile-base, single/dual-arm, rigid/soft-body).

---

[1]† and * indicate equal contribution (in alphabetical order). See Appendix H for contributions.
[2]Project website: https://maniskill2.github.io/
[3]Codes: https://github.com/haosulab/ManiSkill2
[4]Challenge website: https://sapien.ucsd.edu/challenges/maniskill/2022/

## 1 INTRODUCTION

Mastering human-like manipulation skills is a fundamental but challenging problem in Embodied AI, which is at the nexus of vision, learning, and robotics. Remarkably, once humans have learnt to manipulate a category of objects, they are able to manipulate unseen objects (*e.g.*, with different appearances and geometries) of the same category in unseen configurations (*e.g.*, initial poses). We refer such abilities to interact with a great variety of even unseen objects in different configurations as **generalizable manipulation skills**. Generalizable manipulation skills are one of the cornerstones of Embodied AI, which can be composed to tackle long-horizon and complex daily chores (Ahn et al., 2022; Gu et al., 2022). To foster further interdisciplinary and reproducible research on generalizable manipulation skills, it is crucial to build a versatile and public benchmark that focuses on object-level topological and geometric variations as well as practical manipulation challenges.

However, most prior benchmarks are insufficient to support and evaluate progress in learning generalizable manipulation skills. In this work, we present **ManiSkill2**, the next generation of SAPIEN ManiSkill Benchmark (Mu et al., 2021), which extends upon fully simulated dynamics, a large variety of articulated objects, and large-scale demonstrations from the previous version. Moreover, we introduce significant improvements and novel functionalities, as shown below.

**1) A Unified Benchmark for Generic and Generalizable Manipulation Skills:** There does not exist a standard benchmark to measure different algorithms for generic and generalizable manipulation skills. It is largely due to well-known challenges to build realistically simulated environments with diverse assets. Many benchmarks bypass critical challenges as a trade-off, by either adopting abstract grasp (Ehsani et al., 2021; Szot et al., 2021; Srivastava et al., 2022) or including few object-level variations (Zhu et al., 2020; Yu et al., 2020; James et al., 2020). Thus, researchers usually have to make extra efforts to customize environments due to limited functionalities, which in turn makes reproducible comparison difficult. For example, Shridhar et al. (2022) modified 18 tasks in James et al. (2020) to enable few variations in initial states. Besides, some benchmarks are biased towards a single type of manipulation, *e.g.*, 4-DoF manipulation in Yu et al. (2020). To address such pain points, ManiSkill2 includes a total of 20 verified and challenging manipulation task families of multiple types (stationary/mobile-base, single/dual-arm, rigid/soft-body), with over 2000 objects and 4M demonstration frames, to support generic and generalizable manipulation skills. All the tasks are implemented in a unified OpenAI Gym (Brockman et al., 2016) interface with fully-simulated dynamic interaction, supporting multiple observation modes (point cloud, RGBD, privileged state) and multiple controllers. A unified protocol is defined to evaluate a wide range of algorithms (*e.g.*, sense-plan-act, reinforcement and imtation learning) on both seen and unseen assets as well as configurations. In particular, we implement a cloud-based evaluation system to publicly and fairly compare different approaches.

**2) Real-time Soft-body Environments:** When operating in the real world, robots face not only rigid bodies, but many types of soft bodies, such as cloth, water, and soil. Many simulators have supported robotic manipulation with soft body simulation. For example, MuJoCo (Todorov et al., 2012) and Bullet Coumans & Bai (2016–2021) use the finite element method (FEM) to enable the simulation of rope, cloth, and elastic objects. However, FEM-based methods cannot handle large deformation and topological changes, such as scooping flour or cutting dough. Other environments, like SoftGym (Lin et al., 2020) and ThreeDWorld (Gan et al., 2020), are based on Nvidia Flex, which can simulate large deformations, but cannot realistically simulate elasto-plastic material, *e.g.*, clay. PlasticineLab (Huang et al., 2021) deploys the continuum-mechanics-based material point method (MPM), but it lacks the ability to couple with rigid robots, and its simulation and rendering performance have much room for improvement. We have implemented a custom GPU MPM simulator from scratch using Nvidia's Warp (Macklin, 2022) JIT framework and native CUDA for high efficiency and customizability. We further extend Warp's functionality to support more efficient host-device communication. Moreover, we have supported a 2-way dynamics coupling interface that enables any rigid-body simulation framework to interact with the soft bodies, allowing robots and assets in ManiSkill2 to interact with soft-body simulation seamlessly. To our knowledge, ManiSkill2 is the first embodied AI environment to support 2-way coupled rigid-MPM simulation, and also the first to support real-time simulation and rendering of MPM material.

**3) Multi-controller Support and Conversion of Demonstration Action Spaces:** Controllers transform policies' action outputs into motor commands that actuate the robot, which define the action space of a task. Martín-Martín et al. (2019); Zhu et al. (2020) show that the choice of action

space has considerable effects on exploration, robustness and sim2real transferability of RL policies. For example, task-space controllers are widely used for typical pick-and-place tasks, but might be suboptimal compared to joint-space controllers when collision avoidance (Szot et al., 2021) is required. ManiSkill2 supports a wide variety of controllers, *e.g.*, joint-space controllers for motion planning and task-space controllers for teleoperation. A flexible system is also implemented to combine different controllers for different robot components. For instance, it is easy to specify a velocity controller for the base, a task-space position controller for the arm, and a joint-space position controller for the gripper. It differs from Zhu et al. (2020), which only supports setting a holistic controller for all the components. Most importantly, ManiSkill2 embraces a unique functionality to convert the action space of demonstrations to a desired one. It enables us to exploit large-scale demonstrations generated by any approach regardless of controllers.

**4) Fast Visual RL Experiment Support:** Visual RL training demands millions of samples from interaction, which makes performance optimization an important aspect in environment design. Isaac Gym (Makoviychuk et al., 2021) implements a fully GPU-based vectorized simulator, but it lacks an efficient renderer. It also suffers from reduced usability (*e.g.*, difficult to add diverse assets) and functionality (*e.g.*, object contacts are inaccessible). EnvPool (Weng et al., 2022) batches environments by a thread pool to minimize synchronization and improve CPU utilization. Yet its environments need to be implemented in C++, which hinders fast prototyping (*e.g.*, customizing observations and rewards). As a good trade-off between efficiency and customizability, our environments are fully scripted in Python and vectorized by multiple processes. We implement an asynchronous RPC-based render server-client system to optimize throughput and reduce GPU memory usage. We manage to collect samples with an RGBD-input PPO policy at about 2000 FPS [1] with 1 GPU and 16 CPU processors on a regular workstation.

## 2 Building Environments for Generalizable Manipulation Skills

Building high-quality environments demands cross-disciplinary knowledge and expertise, including physical simulation, rendering, robotics, machine learning, software engineering, *etc*. Our workflow highlights a verification-driven iterative development process, which is illustrated in Appendix A.1. Different approaches, including task and motion planning (TAMP), model predictive control (MPC) and reinforcement learning (RL), can be used to generate demonstrations according to characteristics and difficulty of tasks, which verify environments as a byproduct.

### 2.1 Heterogeneous Task Families

ManiSkill2 embraces a heterogeneous collection of 20 task families. A task family represents a family of task variants that share the same objective but are associated with different assets and initial states. For simplicity, we interchangeably use *task* short for *task family*. Distinct types of manipulation tasks are covered: rigid/soft-body, stationary/mobile-base, single/dual-arm. In this section, we briefly describe 4 groups of tasks. More details can be found in Appendix C.

**Soft-body Manipulation**: ManiSkill2 implements 6 soft-body manipulation tasks that require agents to move or deform soft bodies into specified goal states through interaction.
1) *Fill*: filling clay from a bucket into the target beaker;
2) *Hang*: hanging a noodle on the target rod;
3) *Excavate*: scooping up a specific amount of clay and lifting it to a target height;
4) *Pour*: pouring water from a bottle into the target beaker. The final liquid level should match the red line on the beaker.
5) *Pinch*: deforming plasticine from an initial shape into a target shape; target shapes are generated by randomly pinching initial shapes, and are given as RGBD images or point clouds from 4 views.
6) *Write*: writing a target character on clay. Target characters are given as 2D depth maps.

A key challenge of these tasks is to reason how actions influence soft bodies, *e.g.*, estimating displacement quantity or deformations, which will be illustrated in Sec 5.2.

**Precise Peg-in-hole Assembly**: Peg-in-hole assembly is a representative robotic task involving rich contact. We include a curriculum of peg-in-hole assembly tasks that require an agent to place an object into its corresponding slot. Compared to other existing assembly tasks, ours come with two

---

[1] The FPS is reported for rigid-body environments.

noticeable improvements. First, our tasks target high precision (small clearance) at the level of millimeters, as most day-to-day assembly tasks demand. Second, our tasks emphasize the contact-rich insertion process, instead of solely measuring position or rotation errors.

1) *PegInsertionSide*: a single peg-in-hole assembly task inspired by MetaWorld (Yu et al., 2020). It involves an agent picking up a cuboid-shaped peg and inserting it into a hole with a clearance of 3mm on the box. Our task is successful only if half of the peg is inserted, while the counterparts in prior works only require the peg head to approach the surface of the hole.

2) *PlugCharger*: a dual peg-in-hole assembly task inspired by RLBench (James et al., 2020), which involves an agent picking up and plugging a charger into a vertical receptacle. Our assets (the charger and holes on the receptacle) are modeled with realistic sizes, allowing a clearance of 0.5mm, while the counterparts in prior works only examine the proximity of the charger to a predefined position without modeling holes at all.

3) *AssemblingKits*: inspired by Transporter Networks (Zeng et al., 2020), this task involves an agent picking up and inserting a shape into its corresponding slot on a board with 5 slots in total. We devise a programmatic way to carve slots on boards given any specified clearance (*e.g.*, 0.8mm), such that we can generate any number of physically-realistic boards with slots. Note that slots in environments of prior works are visual marks, and there are in fact no holes on boards.

We generate demonstrations for the above tasks through TAMP. This demonstrates that it is feasible to build precise peg-in-hole assembly tasks solvable in simulation environments, without abstractions from prior works.

**Stationary 6-DoF Pick-and-place**: 6-DoF pick-and-place is a widely studied topic in robotics. At the core is grasp pose Mahler et al. (2016); Qin et al. (2020); Sundermeyer et al. (2021). In ManiSkill2, we provide a curriculum of pick-and-place tasks, which all require an agent to pick up an object and move it to a goal specified as a 3D position. The diverse topology and geometric variations among objects call for generalizable grasp pose predictions.

1) *PickCube*: picking up a cube and placing it at a specified goal position;

2) *StackCube*: picking up a cube and placing it on top of another cube. Unlike PickCube, the goal placing position is not explicitly given; instead, it needs to be inferred from observations.

3) *PickSingleYCB*: picking and placing an object from YCB (Calli et al., 2015);

4) *PickSingleEGAD*: picking and placing an object from EGAD (Morrison et al., 2020);

5) *PickClutterYCB*: The task is similar to *PickSingleYCB*, but multiple objects are present in a single scene. The target object is specified by a visible 3D point on its surface.

Our pick-and-place tasks are deliberately designed to be challenging. For example, the goal position is randomly selected within a large workspace ($30 \times 50 \times 50 \, cm^3$). It poses challenges to sense-plan-act pipelines that do not take kinematic constraints into consideration when scoring grasp poses, as certain high-quality grasp poses might not be achievable by the robot.

**Mobile/Stationary Manipulation of Articulated Objects**: We inherit four mobile manipulation tasks from ManiSkill1 (Mu et al., 2021), which are *PushChair*, *MoveBucket*, *OpenCabinetDoor*, and *OpenCabinetDrawer*. We also add a stationary manipulation task, *TurnFaucet*, which uses a stationary arm to turn on faucets of various geometries and topology (details in Appendix C.3).

Besides the above tasks, we have one last task, *AvoidObstacles*, which tests the navigation ability of a stationary arm to avoid a dense collection of obstacles in space while actively sensing the scene.

## 2.2 MULTI-CONTROLLER SUPPORT AND CONVERSION OF DEMONSTRATION ACTION SPACES

The selection of controllers determines the action space. ManiSkill2 supports multiple controllers, *e.g.*, *joint position*, *delta joint position*, *delta end-effector pose*, *etc*. Unless otherwise specified, controllers in ManiSkill2 translate input actions, which are desired configurations (*e.g.*, joint positions or end-effector pose), to joint torques that drive corresponding joint motors to achieve desired actions. For instance, input actions to *joint position*, *delta joint position*, and *delta end-effector pose* controllers are, respectively, desired absolute joint positions, desired joint positions relative to current joint positions, and desired $\mathbb{SE}(3)$ transformations relative to the current end-effector pose. See Appendix B for a full description of all supported controllers.

Demonstrations are generated with one controller, with an associated action space. However, researchers may select an action space that conforms to a task but is different from the original one. Thus, to exploit large-scale demonstrations, it is crucial to convert the original action space

to many different target action spaces while reproducing the kinematic and dynamic processes in demonstrations. Let us consider a pair of environments: a source environment with a *joint position* controller used to generate demonstrations through TAMP, and a target environment with a *delta end-effector pose* controller for Imitation / Reinforcement Learning applications. The objective is to convert the source action $a_{\text{src}}(t)$ at each timestep $t$ to the target action $a_{\text{tgt}}(t)$. By definition, the target action (*delta end-effector pose*) is $a_{\text{tgt}}(t) = \bar{T}_{\text{tgt}}(t) \cdot T_{\text{tgt}}^{-1}(t)$, where $\bar{T}_{\text{tgt}}(t)$ and $T_{\text{tgt}}(t)$ are respectively desired and current end-effector poses in the target environment. To achieve the same dynamic process in the source environment, we need to match $\bar{T}_{\text{tgt}}(t)$ with $\bar{T}_{\text{src}}(t)$, where $\bar{T}_{\text{src}}(t)$ is the desired end-effector pose in the source environment. $\bar{T}_{\text{src}}(t)$ can be computed from the desired joint positions ($a_{\text{src}}(t)$ in this example) through forward kinematics ($FK(\cdot)$). Thus, we have

$$a_{\text{tgt}}(t) = \bar{T}_{\text{tgt}}(t) \cdot T_{\text{tgt}}^{-1}(t) = \bar{T}_{\text{src}}(t) \cdot T_{\text{tgt}}^{-1}(t) = FK(\bar{a}_{\text{src}}(t)) \cdot T_{\text{tgt}}^{-1}(t)$$

Note that our method is closed-loop, as we instantiate a target environment to acquire $T_{\text{tgt}}^{-1}(t)$. For comparison, an open-loop method would use $T_{\text{src}}^{-1}(t)$ and suffers from accumulated execution errors.

## 3 REAL-TIME SOFT BODY SIMULATION AND RENDERING

In this section, we describe our new physical simulator for soft bodies and their dynamic interactions with the existing rigid-body simulation. Our key contributions include: 1) a highly efficient GPU MPM simulator; 2) an effective 2-way dynamic coupling method to support interactions between our soft-body simulator and any rigid-body simulator, in our case, SAPIEN. These features enable us to create the first real-time MPM-based soft-body manipulation environment with 2-way coupling.

**Rigid-soft Coupling**: Our MPM solver is MLS-MPM (Hu et al., 2018) similar to PlasticineLab (Huang et al., 2021), but with a different contact modeling approach, which enables 2-way coupling with external rigid-body simulators. The coupling works by transferring rigid-body poses to the soft-body simulator and transferring soft-body forces to the rigid-body simulator. At the beginning of the simulation, all collision shapes in the external rigid-body simulator are copied to the soft-body simulator. Primitive shapes (box, capsule, etc.) are represented by analytical signed distance functions (SDFs); meshes are converted to SDF volumes, stored as 3D CUDA textures. After each rigid-body simulation step, we copy the poses and velocities of the rigid bodies to the soft-body simulator. During soft-body simulation steps, we evaluate the SDF functions at MPM particle positions and apply penalty forces to the particles, accumulating forces and torques for rigid bodies. In contrast, PlasticineLab evaluates SDF functions on MPM grid nodes, and applies forces to MPM grids. Our simulator supports both methods, but we find applying particle forces produces fewer artifacts such as penetration. After soft-body steps, we read the accumulated forces and torques from the soft-body simulator and apply them to the external rigid-body simulator. This procedure is summarized in Appendix D.1. Despite being a simple 2-way coupling method, it is very flexible, allowing coupling with any rigid-body simulator, so we can introduce anything from a rigid-body simulator into the soft-body environment, including robots and controllers.

**Performance Optimization**: The performance of our soft-body simulator is optimized in 4 aspects. First, the simulator is implemented in Warp, Nvidia's JIT framework to translate Python code to native C++ and CUDA. Therefore, our simulator enjoys performance comparable to C and CUDA. Second, we have optimized the data transfer (*e.g.*, poses and forces in the 2-way coupling) between CPU and GPU by further extending and optimizing the Warp framework; such data transfers are performance bottlenecks in other JIT frameworks such as Taichi (Hu et al., 2019), on which PlasticineLab is based. Third, our environments have much shorter compilation time and startup time compared to PlasticineLab thanks to the proper use of Warp compiler and caching mechanisms. Finally, since the simulator is designed for visual learning environments, we also implement a fast surface rendering algorithm for MPM particles, detailed in Appendix D.2. Detailed simulation parameters and performance are provided in Appendix D.3.

## 4 PARALLELIZING PHYSICAL SIMULATION AND RENDERING

ManiSkill2 aims to be a general and user-friendly framework, with low barriers for customization and minimal limitations. Therefore, we choose Python as our scripting language to model environments, and the open-source, highly flexible SAPIEN (Xiang et al., 2020) as our physical

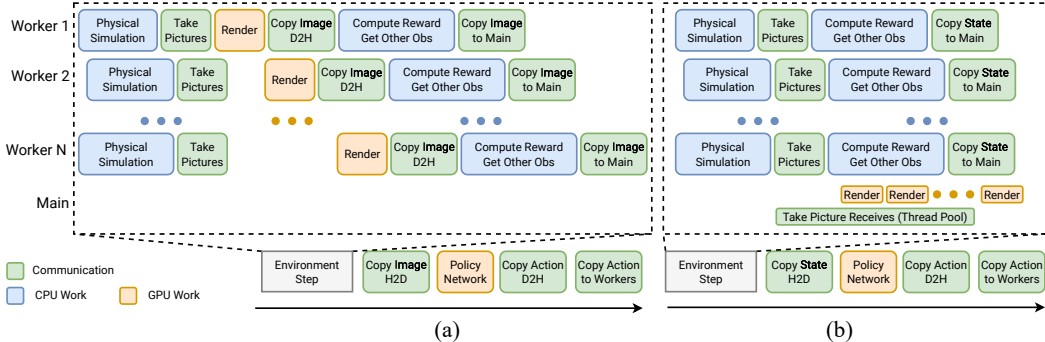

Figure 2: Two pipelines for visual RL sample collection. (a) Sequential pipeline. (b) Our pipeline with asynchronous rendering and render server improves CPU utilization, reduces data transfer, and saves memory.

simulator. The Python language determines that the only way to effectively parallelize execution is through multi-process, which is integrated in common visual RL training libraries (Raffin et al., 2021; Weng et al., 2021). Under the multi-process environment paradigm, we make engineering efforts to enable our environments to surpass previous frameworks by increased throughput and reduced GPU memory usage.

**What are the Goals in Visual-Learning Environment Optimization?** The main goal of performance optimization in a visual-learning environment is to **maximize total throughput**: the number of samples collected from all (parallel) environments, measured in steps (frames) per second. Another goal is to **reduce GPU memory usage**. Typical Python-based multi-process environments are wasteful in GPU memory usage, since each process has to maintain a full copy of GPU resources, *e.g.*, meshes and textures. Given a fixed GPU memory budget, less GPU memory spent on rendering resources means more memory can be allocated for neural networks.

**Asynchronous Rendering and Server-based Renderer**: Fig. 2(a) illustrates a typical pipeline (sequential simulation and rendering) to collect samples from multi-process environments. It includes the following stages: (1) do physical simulation on worker processes; (2) take pictures (update renderer GPU states and submit draw calls); (3) wait for GPU render to finish; (4) copy image observations to CPU; (5) compute rewards and get other observations (e.g., robot proprioceptive info); (6) copy images to the main python process and synchronize; (7) copy these images to GPU for policy learning; (8) forward the policy network on GPU; (9) copy output actions from the GPU to the simulation worker processes.

We observe that the CPU is idling during GPU rendering (stage 3), while reward computation (stage 5) often does not rely on rendering results. Thus, we can increase CPU utilization by starting reward computation immediately after stage 2. We refer to this technique as **asynchronous rendering**.

Another observation is that images are copied from GPU to CPU on each process, passed to the main python process, and uploaded to the GPU again. It would be ideal if we can keep the data on GPU at all times. Our solution is to use a **render server**, which starts a thread pool on the main Python process and executes rendering requests from simulation worker processes, summarized in figure 2(b). The render server eliminates GPU-CPU-GPU image copies, thereby reducing data transfer time. It allows GPU resources to share across any number of environments, thereby significantly reducing memory usage. It enables communication over network, thereby having the potential to simulate and render on multiple machines. It requires minimal API changes – the only change to the original code base is to receive images from the render server. It also has additional benefits in software development with Nvidia GPU, which we detail in Appendix E.2.

**Comparison**: We compare the throughput of ManiSkill2 with 3 other framework that support visual RL: Habitat 2.0 (Szot et al., 2021), RoboSuite 1.3 (Zhu et al., 2020), and Isaac Gym (Makoviychuk et al., 2021). We build a *PickCube* environment in all simulators. We use similar physical simulation parameters (500Hz simulation frequency and 20Hz control frequency[2]), and we use the GPU

---

[2]A fair comparison of different frameworks is still challenging as their simulation and rendering differ in fidelity. We try our best to match the simulation parameters among the frameworks.

|  | ManiSkill2 Server | ManiSkill2 Sync | Habitat 2.0 | RoboSuite 1.3 | Isaac Gym | #Envs | ManiSkill2 Server | Habitat 2.0 |
|---|---|---|---|---|---|---|---|---|
| Total FPS (rand. action) | **2487±24** | 942±19 | 1275±10 | 924±3 | 865±35 | 4 | 4.9G | 6.4G |
| Total FPS (nature CNN) | **2532±63** | 931±4 | 1224±13 | 894±15 | 835±5 | 8 | 5.1G | 12.9G |
| Optimal #Envs | 64 | 32 | 64 | 32 | 512 | 64 | 5.8G | (OOM) |

|  |  |
|---|---|
| (a) | (b) |

Table 1: (a) Comparison of sample collection speed (FPS) on *PickCube* across different frameworks. (b) GPU memory usage on multi-process environments with 74 YCB objects each.

pipeline for rendering. Images have resolution 128x128. All frameworks are given a computation budget of 16 CPU cores (logical processors) of an Intel i9-9960X CPU with 128G memory and 1 RTX Titan GPU with 24G memory. We test with random actions and also with a randomly initialized nature CNN (Mnih et al., 2015) as policy network. We test all frameworks on 16, 32, 64, 128, 256, 512 parallel environments and report the highest performance. Results are shown in Table 1(a) (more details in Appendix E.3). An interesting observation is that our environment performs the best when the number of parallel environments exceeds the number of CPU cores. We conjecture that this is because execution on CPU and GPU are naturally efficiently interleaved through OS or driver-level schedulers when the requested computation far exceeds the available resources.

Additionally, we demonstrate the advantage of memory sharing thanks to our render server. We extend the *PickClutterYCB* environment to include 74 YCB objects per scene, and create the same setting in Habitat 2.0. As shown in Table 1(b), even though we enable GPU texture compression for Habitat and use regular textures for ManiSkill2, the memory usage of Habitat grows linearly as the number of environments increases, while ManiSkill2 requires very little memory to create additional environments since all meshes and textures are shared across environments.

## 5 APPLICATIONS

In this section, we show how ManiSkill2 supports widespread applications, including sense-plan-act frameworks and imitation / reinforcement learning. In this section, for tasks that have asset variations, we report results on training objects. Results on held-out objects are presented in Appendix F. Besides, we demonstrate that policies trained in ManiSkill2 have the potential to be directly deployed in the real world.

### 5.1 SENSE-PLAN-ACT

Sense-plan-act (SPA) is a classical framework in robotics. Typically, a SPA method first leverages perception algorithms to build a world model from observations, then plans a sequence of actions through motion planning, and finally execute the plan. However, SPA methods are usually open-loop and limited by motion planning, since perception and planning are independently optimized. In this section, we experiment with two perception algorithms, Contact-GraspNet (Sundermeyer et al., 2021) and Transporter Networks (Zeng et al., 2020).

**Contact-GraspNet for *PickSingleYCB***: The SPA solution for *PickSingleYCB* is implemented as follows. First, we use Contact-GraspNet (CGN) to predict potential grasp poses along with confidence scores given the observed partial point cloud. We use the released CGN model pre-trained on ACRONYM (Eppner et al., 2021). Next, we start with the predicted grasp pose with the highest score, and try to generate a plan to achieve it by motion planning. If no plan is found, the grasp pose with the next highest score is attempted until a valid plan is found. Then, the agent executes the plan to reach the desired grasp pose and closes its grippers to grasp the object. Finally, another plan is generated and executed to move the gripper to the goal position.

For each of the 74 objects, we conduct 5 trials (different initial states). The task succeeds if the object's center of mass is within 2.5cm of the goal. The success rate is 43.24%. There are two main failure modes: 1) predicted poses are of low confidence (27.03% of trials), especially for objects (*e.g.*, spoon and fork) that are small and thin; 2) predicted poses are of low grasp quality or unreachable (29.73% of trials), but with high confidence. See Appendix F.1 for examples.

**Transporter Network for *AssemblingKits***: We benchmark Transporter Networks (TPN) on our *AssemblingKits*. The original success metric (pose accuracy) used in TPN is whether the peg is

| Obs. Mode | PickCube | StackCube | Fill | Hang | Excavate | Pour | Pinch | Write |
|---|---|---|---|---|---|---|---|---|
| Point Cloud | $0.22 \pm 0.06$ | $0.04 \pm 0.02$ | $0.45 \pm 0.04$ | $0.35 \pm 0.15$ | $0.08 \pm 0.04$ | $0.02 \pm 0.02$ | $0.00 \pm 0.00$ | $0.00 \pm 0.00$ |
| RGBD | $0.01 \pm 0.02$ | $0.00 \pm 0.00$ | $0.62 \pm 0.07$ | $0.20 \pm 0.12$ | $0.21 \pm 0.04$ | $0.00 \pm 0.00$ | $0.00 \pm 0.00$ | $0.00 \pm 0.00$ |

Table 2: Mean and standard deviation of the success rate of behavior cloning on rigid-body and soft-body tasks. For rigid-body assembly tasks not shown in the table, success rate is 0.

| Obs. Mode | PickCube | StackCube | PickSingleYCB | PegInsSide | PlugCharger | AssemblingKits | TurnFaucet | AvoidObstacles |
|---|---|---|---|---|---|---|---|---|
| Point Cloud | $0.94 \pm 0.03$ | $0.95 \pm 0.02$ | $0.51 \pm 0.05$ | $0.01 \pm 0.01$ | $0.01 \pm 0.02$ | $0.00 \pm 0.00$ | $0.04 \pm 0.03$ | $0.00 \pm 0.00$ |
| RGBD | $0.91 \pm 0.05$ | $0.87 \pm 0.04$ | $0.18 \pm 0.07$ | $0.01 \pm 0.01$ | $0.01 \pm 0.01$ | $0.00 \pm 0.00$ | $0.03 \pm 0.03$ | $0.00 \pm 0.00$ |

Table 3: Mean and standard deviation of success rates of DAPG+PPO on rigid-body tasks. Training budget is 25M time steps.

placed within 1cm and 15 degrees of the goal pose. Note that our version requires pieces to actually fit into holes, and thus our success metric is much stricter. We train TPN from scratch with image data sampled from training configurations using two cameras, a base camera and a hand camera. To address our high-precision success criterion, we increase the number of bins for rotation prediction from 36 in the original work to 144. During evaluation, we employ motion planning to move the gripper to the predicted pick position, grasp the peg, then generate another plan to move the peg to the predicted goal pose and drop it into the hole. The success rate over 100 trials is 18% following our success metric, and 99% following the pose accuracy metric of (Zeng et al., 2020). See Appendix F.2 for more details.

## 5.2 IMITATION & REINFORCEMENT LEARNING WITH DEMONSTRATIONS

For the following experiments, unless specified otherwise, we use the *delta end-effector pose* controller for rigid-body environments and the *delta joint position* controller for soft-body environments, and we translate demonstrations accordingly using the approach in Sec.2.2. Visual observations are captured from a base camera and a hand camera. For RGBD-based agents, we use IMPALA (Espeholt et al., 2018) as the visual backbone. For point cloud-based agents, we use PointNet (Qi et al., 2017) as the visual backbone. In addition, we transform input point clouds into the end-effector frame, and for environments where goal positions are given (PickCube and PickSingleYCB), we randomly sample 50 green points around the goal position to serve as visual cues and concatenate them to the input point cloud. We run 5 trials for each experiment and report the mean and standard deviation of success rates. Further details are presented in Appendix F.3.

**Imitation Learning**: We benchmark imitation learning (IL) with behavior cloning (BC) on our rigid and soft-body tasks. All models are trained for 50k gradient steps with batch size 256 and evaluated on test configurations.

Results are shown in Table 2. For rigid-body tasks, we observe low or zero success rates. This suggests that BC is insufficient to tackle many crucial challenges from our benchmark, such as precise control in assembly and large asset variations. For soft-body tasks, we observe that it is difficult for BC agents to precisely estimate action influence on soft body properties (e.g. displacement quantity or deformation). Specifically, agents perform poorly on *Excavate* and *Pour*, as *Excavate* is successful only if a specified amount of clay is scooped up and *Pour* is successful when the final liquid level accurately matches a target line. On the other hand, for *Fill* and *Hang*, which do not have such precision requirements (for *Fill*, the task is successful as long as the amount of particles in the beaker is larger than a threshold), the success rates are much higher. In addition, we observe that BC agents cannot well utilize target shapes to guide precise soft body deformation, as they are never successful on *Pinch* or *Write*. See Appendix F.5 for further analysis.

**RL with Demonstrations**: We benchmark demonstration-based online reinforcement learning by augmenting Proximal Policy Gradient (PPO) (Schulman et al., 2017) objective with the demonstration objective from Demonstration-Augmented Policy Gradient (DAPG) (Rajeswaran et al., 2017). Our implementation is similar to Jia et al. (2022), and further details are presented in Appendix F.3. We train all agents from scratch for 25 million time steps. Due to limited computation resources, we only report results on rigid-body environments.

Figure 3: Left: Simulation and real world setup for *PickCube*. Right: Motion planning execution results for *Pinch* in simulation and in the real world: (a) initial state; (b) letter "M"; (c) letter "S".

Results are shown in Table 3. We observe that for pick-and-place tasks, point cloud-based agents perform significantly better than RGBD-based agents. Notably, on *PickSingleYCB*, the success rate is even higher than Contact-GraspNet with motion planning. This demonstrates the potential of obtaining a single agent capable of performing general pick-and-place across diverse object geometries through online training. We also further examine factors that influence point cloud-based manipulation learning in Appendix F.6. However, for all other tasks, notably the assembly tasks that require high precision, the success rates are near zero. In Appendix F.7, we show that if we increase the clearance of the assembly tasks and decrease their difficulty, agents can achieve much higher performance. This suggests that existing RL algorithms might have been insufficient yet to perform highly precise controls, and our benchmark poses meaningful challenges for the community.

In addition, we examine the influence of controllers for agent learning, and we perform ablation experiments on *PickSingleYCB* using point cloud-based agents. When we replace the *delta end-effector pose* controller in the above experiments with the *delta joint position* controller, the success rate falls to 0.22±0.18. The profound impact of controllers demonstrates the necessity and significance of our multi-controller conversion system.

## 5.3 SIM2REAL

ManiSkill2 features fully-simulated dynamic interaction for rigid-body and soft-body. Thus, policies trained in ManiSkill2 have the potential to be directly deployed in the real world.

***PickCube***: We train a visual RL policy on *PickCube* and evaluate its transferability to the real world. The setup is shown in Fig. 3-L, which consists of an Intel RealSense D415 depth sensor, a 7DoF ROKAE xMate3Pro robot arm, a Robotiq 2F-140 2-finger gripper, and the target cube. We first acquire the intrinsic parameters for the real depth sensor and its pose relative to the robot base. We then build a simulation environment aligned with the real environment. We train a point cloud-based policy for 10M time steps, where the policy input consists of robot proprioceptive information (joint position and end-effector pose) and uncolored point cloud backprojected from the depth map. The success rate in simulation is 91.0%. We finally directly evaluate this policy in the real world 50 times with different initial states, where we obtain 60.0% success rate. We conjecture that the performance drop is caused by the domain gap in depth maps, as we only used Gaussian noise and random pixel dropout as data augmentation during simulation policy training.

***Pinch***: We further evaluate the fidelity of our soft-body simulation by executing the same action sequence generated by motion planning in simulation and in the real world and comparing the final plasticine shapes. Results are shown in Fig. 3-R, which demonstrates that our 2-way coupled rigid-MPM simulation is able to reasonably simulate plasticine's deformation caused by multiple grasps.

## 6 CONCLUSION

To summarize, ManiSkill2 is a unified and accessible benchmark for generic and generalizable manipulation skills, providing 20 manipulation task families, over 2000 objects, and over 4M demonstration frames. It features highly efficient rigid-body simulation and rendering system for sample collection to train RL agents, real-time MPM-based soft-body environments, and versatile multi-controller conversion support. We have demonstrated its applications in benchmarking sense-plan-act and imitation / reinforcement learning algorithms, and we show that learned policies on ManiSkill2 have the potential to transfer to the real world.

REPRODUCIBLITY STATEMENT

Our Benchmark, algorithms, and applications are fully open source. Specifically, we open source the ManiSkill benchmark as well as all simulators and renderers used to build it. We open source all code for demonstration generation and controller / action space conversion. We open source the entire learning framework used to train Imitation / Reinforcement Learning and Sense-Plan-Act algorithms. We release all training assets used in ManiSkill2. Since we will hold a public challenge based on ManiSkill2, code related to asset generation and cloud-based evaluation service cannot be released for fairness. Nonetheless, all results in this work are reproducible, and we welcome researchers to participate in our challenge.

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

## A  SYSTEM DESIGN FOR DEVELOPMENT AND EVALUATION

### A.1  VERIFICATION-DRIVEN ITERATIVE DEVELOPMENT

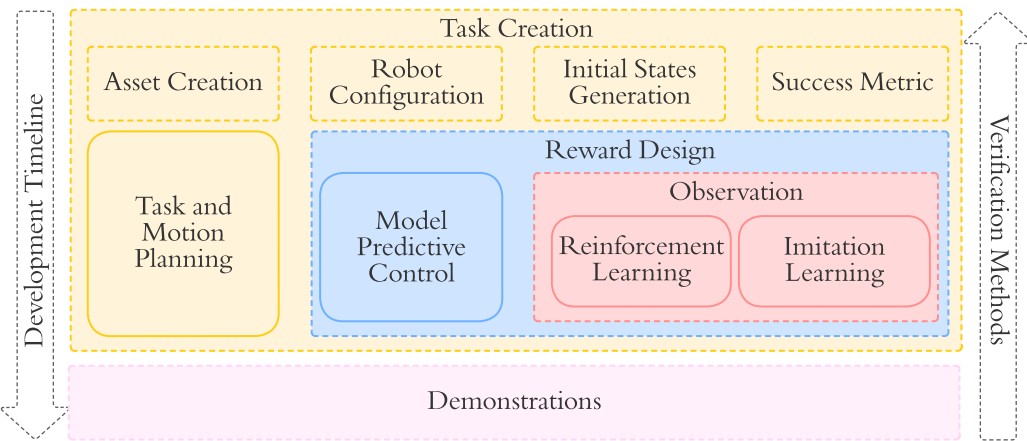

Figure 4: The workflow to build environments for generalizable manipulation skills.

Our workflow, following agile software development (Beck et al., 2001), highlights a verification-driven iterative development process. Different approaches can be used to generate demonstrations according to characteristics and difficulties of tasks, which verifies environments as a byproduct. The workflow is also designed to be scalable and affordable to continuous integration of assets and tasks. Fig 4 illustrates the workflow.

Our workflow consists of 3 stages: task creation, reward design, and observation configuration. The first stage focuses on building task essentials, including creating assets, (*e.g.*, convex decomposition (Wei et al., 2022), texture baking), configuring robots, generating initial states, and defining success metrics. The second stage aims at prototyping shaped reward functions. The reward function is a requisite for methods like Model-Predictive Control (MPC) and RL. The third stage addresses observation spaces. For instance, camera parameters and placements need to be tailored for tasks so that observations contain adequate information. To collect demonstrations and verify environments, we employ one of or a mixture of 3 complementary approaches: task and motion planning (TAMP), MPC, and RL.

Different methods have their own advantages and disadvantages. TAMP is free of crafting reward, and is suitable for many stationary manipulation tasks like pick-and-place, but shows difficulty when tackling underactuated systems (e.g. pushing chairs and moving buckets in Mu et al. (2021)). MPC is able to search solutions to difficult tasks given well-designed shaped rewards without training or observations. However, it is non-trivial to design a universal shaped reward for a variety of objects. RL requires additional training and hyperparameter tuning, but is more scalable than MPC during inference.

### A.2  CLOUD BASED EVALUATION SYSTEM

To allow the community to evaluate and benchmark together we build and provide a simple cloud based evaluation system. Users can register accounts and enter the benchmark and begin making submissions that solve our various tasks.

A key feature of the evaluation system is flexibility. This is increasingly important as the number of unique approaches from heuristics, motion planners, and end-to-end RL solutions grow. Evaluation systems need the flexibility to allow all kinds of approaches to be benchmarked. To this end, we do the following.

User submissions are in the form of docker images, allowing users to install any code and save any models necessary to solve various tasks. This makes programming on the user's side very flexible. The user only needs to provide a function that accepts observations and returns actions.

We further give users the flexibility to configure the evaluation environment to suit their needs before running the evaluation. For example, users can define a configuration function in their solution that sets the observation mode (e.g. RGB-D or Point Cloud), as well as controllers (e.g. *delta end-effector pose* or *joint position*).

To benchmark a submitted docker image, we simply pull it to our server and run the evaluation code that loads the user's solution. The results of the evaluation are then submitted to a database and displayed on a pubic benchmark.

# B    DETAILS OF OUR COMPREHENSIVE CONTROLLER SUITE

Controllers are interfaces between policies and robots. Policies output actions to controllers, and controllers convert actions to control signals to drive the robot joints. In ManiSkill2, the default robot being controlled is the Franka Emika, also known as Panda. The degree of freedom (DoF) of a single Panda arm is 7.

## B.1    TERMINOLOGY

1. **Fixed Joint**: A joint that cannot be controlled. The degree of freedom (DoF) is 0.

2. **qpos** ($q$): Controllable joint positions.

3. **qvel** ($\dot{q}$): Controllable joint velocities.

4. **Target Joint Positions** ($\bar{q}$): Target positions of motors that drive each joint.

5. **Target Joint Velocities** ($\dot{\bar{q}}$): Target velocities of motors that drive each joint.

6. **End-effector Position** ($p$): The position of an end-effector.

7. **End-effector Rotation** ($R$): The rotation of an end-effector.

8. **End-effector Target Position** ($\bar{p}$): The target position of an end-effector.

9. **End-effector Target Rotation** ($\bar{R}$): The target rotation of an end-effector.

10. **PD Controller**: Control loop based on $\tau(t) = K_p(\bar{q}(t) - q(t)) + K_d(\dot{\bar{q}}(t) - \dot{q}(t))$. $K_p$ (stiffness) and $K_d$ (damping) are hyperparameters. $\tau(t)$ denotes the torque (generalized force) of the motors.

11. **Augmented PD Controller**: Augmented PD controller: Passive forces (like gravity) are compensated for the PD controller.

12. **Action** ($a$): Input to the controllers, and also output of the policy.

13. **Tool Center Point** (TCP): TCP is a user defined coordinate frame, often relatively fixed to the robot end effector. For example, in our case, if the robot uses a two-finger gripper, TCP is defined at the center point between the gripper's two fingers.

## B.2    TARGET VS. NON-TARGET CONTROLLERS

In our controller implementation, we have the notion of "target" and "non-target" controllers. For example, when we say *delta end-effector pose* controller, the new desired pose is specified relative to the current end-effector pose. In contrast, if we say *target delta end-effector pose* controller, the new desired pose is specified relative to the previous desired pose. Please read the next section for their formal definitions.

## B.3    NORMALIZED ACTION SPACE

We design the action space of controllers to conform to the preferences of users. As RL users generally prefer normalized action space, most controllers listed below will have a normalized action space $[-1, 1]$, with a few exceptions listed individually.

### B.4 DETAILS OF CONTROLLERS

#### B.4.1 ARM CONTROLLERS

1. *arm_pd_joint_pos* (7-dim, unnormalized): $a_t = \bar{q}_t$. The target joint velocities $\dot{q}_t$ are always 0. As this controller is suitable for motion planning, the action space of this controller is not normalized.

2. *arm_pd_joint_delta_pos* (7-dim): $a_t = \bar{q}_t - q_{t-1}$.

3. *arm_pd_joint_target_delta_pos* (7-dim): $a_t = \bar{q}_t - \bar{q}_{t-1}$.

4. *arm_pd_ee_delta_pos* (3-dim): $a_t = \bar{p}_t - p_{t-1}$, where $p_t$ is the position of the end-effector at timestep $t$. The controller then internally computes the target joint positions of the robot: $q_t = IK(\bar{p}_t, \bar{R}_t)$, where $IK(\cdot, \cdot)$ computes the joint positions from the end-effector's position and rotation before sending the joint positions to the PD controller. Note that this controller only controls the position, but not the rotation, of the end-effector.

5. *arm_pd_ee_target_delta_pos* (3-dim): $a_t = \bar{p}_t - \bar{p}_{t-1}$

6. *arm_pd_ee_delta_pose* (6-dim): This controller is very similar to the previous *arm_pd_ee_delta_pos* controller with the addition of end-effector's rotation being passed in as input. Thus $\bar{T}_t = T_a \cdot T_{t-1}$, where $\bar{T}$ is the target transformation of the end-effector, and $T_a$ is the delta pose induced by the 3D position and 3D rotation (represented as compact axis-angle) of the action.

7. *arm_pd_ee_target_delta_pose* (6-dim): $\bar{T}_t = T_a \cdot \bar{T}_{t-1}$.

8. *arm_pd_joint_vel* (7-dim): $a_t = \bar{\dot{q}}_t$. The stiffness value $K_p$ of this controller is always set to 0.

9. *arm_pd_joint_pos_vel* (14-dim): An extension of *arm_pd_joint_pos* that supports target velocities input.

10. *arm_pd_joint_delta_pos_vel* (14-dim): The delta variant of the *arm_pd_joint_pos_vel* controller.

#### B.4.2 GRIPPER CONTROLLER (1-DIM)

We use joint position control for the gripper, and we force the two gripper fingers to have the same target position.

### B.5 EFFECTIVENESS OF CONVERSION OF DEMONSTRATION ACTION SPACES

In this section, we exemplify the success rate of our demonstration conversion method by converting from the *arm_pd_joint_pos* controller to the *arm_pd_ee_delta_pose* controller. A demonstration is converted successfully if, following the same trajectory initialization and the converted actions, the task is successful at the last time step. We experiment on 4 representative tasks: *PickSingleYCB*, *AssemblingKits*, *TurnFaucet*, *Write*. For each task, we select 100 demonstrations randomly. The success rates for *PickSingleYCB*, *AssemblingKits*, *Write*, *TurnFaucet* are 99%, 98%, 100%, and 80%, respectively. Note that our policy to generate demonstrations for *TurnFaucet* involves rich and inconsistent contact between the end-effector and the faucet handle (i.e., our policy uses the gripper to push the handle, rather than grasping and rotating it, in which case force closure can lead to more consistent contact). Such polices can be sensitive to accumulated errors during execution, which can result in task failure, although the actions converted by our demonstration conversion method attempt to reproduce motion faithfully.

## C DETAILS OF OBSERVATIONS, TASK FAMILIES, DEMONSTRATIONS AND EVALUATION PROTOCOLS

Unless otherwise noted, all demonstrations are generated through TAMP. **For evaluation, we employ a two-stage setup. Final result is the average result from the two stages.**

### C.1 SUPPORTED OBSERVATION MODES

We support most observation modes found in OpenAI gym (Brockman et al., 2016). The details of each observation mode are listed below.

1. *state_dict*: Returns a dictionary of states that contains robot proprioceptive information, ground truth object information (such as object poses), and task-specific goal information (if given). Visual observations (images and point clouds) are **not** included.

2. *state*: Returns the flattened version of a *state_dict*.

3. *rgbd*: Returns rendered RGBD images from all cameras, along with robot proprioceptive information and task-specific goal information (if given).

4. *rgbd_robot_seg*: On top of *rgbd*, returns ground truth segmentation masks for the robot joints.

5. *pointcloud*: Returns a fused point cloud from all cameras, along with robot proprioceptive information and task-specific goal information (if given).

6. *pointcloud_robot_seg*: On top of *pointcloud*, returns ground truth segmentation masks for the robot joints.

Here, the robot proprioceptive information includes joint positions, joint velocities, the pose of the robot base, along with the pose of the gripper's tool center point if the robot uses a two-finger gripper.

Note that different from the previous version of ManiSkill, ManiSkill2 does not include ground-truth segmentation in the default observation modes (*rgbd* or *pointcloud*). All visual-based experiments in this paper do not leverage such privileged information. For example, to specify which link of a faucet should be manipulated in *TurnFaucet*, we use its initial position instead of a ground-truth segmentation mask. Besides, we also support observations modes (*rgbd+robot_seg*, *pointcloud+robot_seg*) to provide the segmentation masks of robot links, which facilitates robotic applications and can be obtained in the real world using the robot proprioceptive information.

### C.2 PICK-AND-PLACE

**PickCube**

- Objective: Pick up a cube and move it to a goal position.
- Success Metric: The cube is within 2.5 cm of the goal position, and the robot is static.
- Demonstration Format: 1,000 successful trajectories.
- Evaluation Protocol: 100 episodes with different initial joint positions of the robot and initial cube pose for each of stage 1 and stage 2.
- Task-specific Extra Observations: 3D goal position of the cube.

**StackCube**

- Objective: Pick up a red cube and place it onto a green one.
- Success Metric: The red cube is placed on top of the green one stably and it is not grasped.
- Demonstration Format: 1,000 successful trajectories.
- Evaluation Protocol: 100 episodes with different initial joint positions of the robot and initial poses of both cubes for each of stage 1 and stage 2.
- Task-specific Extra Observations: None.

**PickSingleYCB**

- Objective: Pick up a YCB object and move it to a goal position.
- Success Metric: The object is within 2.5 cm of the goal position, and the robot is static.
- Demonstration Format: 100 successful trajectories for each of the 74 YCB objects.

- Evaluation Protocol: In addition to the training objects, we also use another confidential set of 40 objects from other sources as the test object set. For the two evaluation stages in total, for each object in the training set, we test 5 episodes with different seeds. For each object in the test set, we test 10 episodes with different seeds. Half of the objects are evaluated in each stage.
- Task-specific Extra Observations: 3D goal position of the object.

**PickSingleEGAD**

- Objective: Pick up an EGAD object and move it to a goal position. The color for the EGAD object is randomized.
- Success Metric: The object is within 2.5 cm of the goal position, and the robot is static.
- Demonstration Format: 5 trajectories for each of the 1,600 training objects sampled from EGAD. For certain objects where it's difficult to apply TAMP, we might provide less than 5 trajectories.
- Evaluation Protocol: For this task, we have held out a portion of the EGAD dataset. This held-out test dataset consists of 150 objects. During evaluation, in each stage, we evaluate 1 trajectory for each of the 150 objects sampled from the training dataset and 2 trajectories for each of the 75 objects sampled from the held-out test dataset.
- Task-specific Extra Observations: 3D goal position of the object.

**PickClutterYCB**

- Objective: Pick up an object from a clutter of 4-8 YCB objects.
- Success Metric: The object is within 2.5 cm of the goal position, and the robot is static.
- Demonstration Format: A total of 4986 trajectories from the training object set.
- Evaluation Protocol: In addition to the training objects, we also use another confidential set of 40 objects from other sources as the test object set. For each evaluation stage, we test 100 episode configurations on the training object set and on the test object set.
- Task-specific Extra Observations: 3D position of the object to pick up, and 3D position of the goal.

## C.3 ASSEMBLY

**AssemblingKits**

- Objective: Insert an object into the corresponding slot on a plate.
- Success Metric: An object must fully fit into its slot, which must simultaneously satisfy 3 criteria: (1) height of the object center is within 3mm of the height of the plate; (2) rotation error is within 4 degrees; (3) position error is within 2cm.
- Demonstration Format: We provide 1,720 trajectories in total. These trajectories are generated from over 300 kit configurations and 20 training shapes.
- Evaluation Protocol: This task has a held-out test dataset for evaluation. The test dataset features 20 shapes that are similar to the shapes in the training set. We provide samples for test assets in Fig. 5. In each evaluation stage, we evaluate on 100 sampled training episode configurations and 100 sampled test dataset configurations.
- Task-specific Extra Observations: 3D initial and goal position of the object to be placed.

**PegInsertionSide**

- Objective: Insert a peg into the horizontal hole in a box.
- Success Metric: Half of the peg is inserted into the hole.
- Demonstration Format: 1,000 successful trajectories.
- Evaluation Protocol: 100 episodes with different initial joint positions of the robot, initial poses of the peg and box, the position and size of the hole for each of stage 1 and stage 2.
- Task-specific Extra Observations: None.

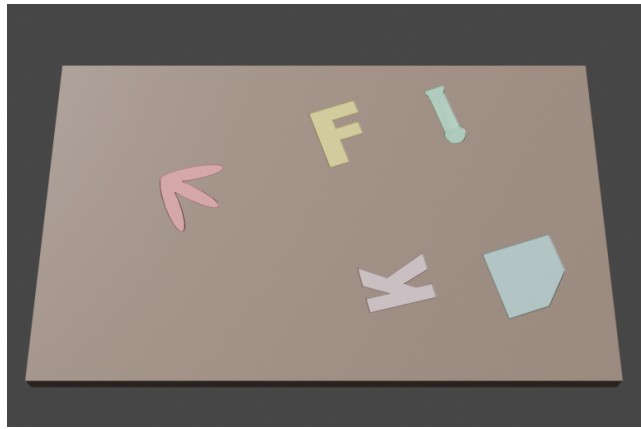

Figure 5: A sample plate with test assets.

**PlugCharger**

- Objective: Plug a charger into a wall receptacle.
- Success Metric: The charger is fully inserted into the receptacle.
- Demonstration Format: 1,000 successful trajectories.
- Evaluation Protocol: 100 episodes with different initial joint positions of the robot, initial poses of the charger and wall for each of stage 1 and stage 2.
- Task-specific Extra Observations: None.

C.4    MISCELLANEOUS TASKS

**AvoidObstacles**

- Objective: Navigate the robot arm through a region of dense obstacles and move the end-effector to a goal pose. The shape and color of dense obstacles are randomized.
- Success Metric: The end-effector pose is within 2.5 cm and 15 degrees of the goal pose.
- Demonstration Format: 1976 trajectories for different layouts.
- Evaluation Protocol: 100 episodes with different layouts of obstacles for each of stage 1 and stage 2.
- Task-specific Extra Observations: The goal pose of the end-effector.

**TurnFaucet**

- Objective: Turn on a faucet by rotating its handle.
- Success Metric: The faucet handle has been turned past a target angular distance.
- Demonstration Format: For most faucet models, we provide 100 trajectories per asset. For approximately 15 of the 60 training models were TAMP cannot find a solution, demonstrations are generated through MPC-CEM using our designed rewards.
- Evaluation Protocol: This task has a held-out test object set. For the two evaluation stages in total, we evaluate 5 episodes for each of the 60 training objects and 17 episodes for each of the 18 test objects. Half of the objects are evaluated in each stage.
- Task-specific Extra Observations: The remaining angular distance to rotate the handle, the target handle position (since there can be multiple handles in a single faucet), and the direction to rotate the handle specified as 3D joint axis.

C.5    SOFT-BODY MANIPULATION

**Fill**

- Objective: Fill clay from a bucket into the target beaker.
- Success Metric: The amount of clay inside the target beaker $> 90\%$; soft body velocity $< 0.05$.
- Demonstration Format: 200 successful trajectories generated through motion planning.
- Evaluation Protocol: 100 episodes with different initial rotations of the bucket and initial positions of the beaker for each of stage 1 and stage 2.
- Task-specific Extra Observations: Beaker position.

**Hang**

- Objective: Hang a noodle on a target rod.
- Success Metric: Part of the noodle is higher than the rod; two ends of the noodle are on different sides of the rod; the noodle is not touching the ground; the gripper is open; soft body velocity $< 0.05$.
- Demonstration Format: 200 successful trajectories generated through motion planning.
- Evaluation Protocol: 100 episodes with different initial positions of the gripper and rod poses for each of stage 1 and stage 2.
- Task-specific Extra Observations: Rod position.

**Excavate**

- Objective: Lift a specific amount of clay to a target height.
- Success Metric: The amount of lifted clay must be within a given range; the lifted clay is higher than a specific height; fewer than 20 clay particles are spilled on the ground; soft body velocity $< 0.05$.
- Demonstration Format: 200 successful trajectories generated through motion planning.
- Evaluation Protocol: 100 episodes with different bucket poses and initial heightmaps of clay for each of stage 1 and stage 2.
- Task-specific Extra Observations: Target clay amount.

**Pour**

- Objective: Pour liquid from a bottle into a beaker.
- Success Metric: The liquid level in the beaker is within 4mm of the red line; the spilled water is fewer than 100 particles; the bottle returns to the upright position in the end; robot arm velocity $< 0.05$.
- Demonstration Format: 200 successful trajectories generated through motion planning.
- Evaluation Protocol: 100 episodes with different bottle positions, the level of water in the bottle, and beaker positions for each of stage 1 and stage 2.
- Task-specific Extra Observations: Red line height.

**Pinch**

- Objective: Deform plasticine into a target shape.
- Success Metric: The Chamfer distance between the current plasticine and the target shape is less than $0.3t$, where $t$ is the Chamfer distance between the initial shape and target shape.
- Demonstration Format: 1556 successful trajectories generated through heuristic motion planning. Different trajectories correspond to different target shapes.
- Evaluation Protocol: 50 episodes with different target shapes for each of stage 1 and stage 2.
- Task-specific Extra Observations: RGBD / point cloud observations of the target plasticine from 4 different views.

**Write**

- Objective: Write a given character on clay. The target character is randomly sampled from an alphabet of over 50 characters.
- Success Metric: The IoU (Intersection over Union) between the current pattern and the target character is larger than 0.8.
- Demonstration Format: 200 successful trajectories generated through heuristic motion planning.
- Evaluation Protocol: 50 episodes with different target characters for each of stage 1 and stage 2.
- Task-specific Extra Observations: The depth map of the target character.

## C.6    MOBILE MANIPULATION

**OpenCabinetDrawer**

- Objective: A single-arm mobile robot needs to open a designated target drawer on a cabinet. The friction and damping parameters for the drawer joints are randomized.
- Success Metric: The target drawer has been opened to at least 90% of the maximum range, and the target drawer is static.
- Demonstration Format: 300 trajectories for each target drawer in the training object set. The training object set consists of 25 cabinets. Each cabinet could contain multiple drawers.
- Evaluation Protocol: This task has a held-out test object set (10 unseen cabinets). In the first stage, we evaluate 250 trajectories in total. Among these 250 trajectories, 125 levels are evenly distributed over 5 unseen objects in the test set, and the other 125 levels are evenly distributed over all objects in the training set. In the second stage, we evaluate another 250 trajectories. Similarly, 125 levels come from the training set and the other 125 levels from the 5 other unseen objects in the test set (different from the 5 test objects in stage 1).
- Task-specific Extra Observations: Since one cabinet can contain several drawers, we specify the target drawer by its initial center of mass.

**OpenCabinetDoor**

- Objective: A single-arm mobile robot needs to open a designated target door on a cabinet. The friction and damping parameters for the door joints are randomized.
- Success Metric: The target door has been opened to at least 90% of the maximum range, and the target door is static.
- Demonstration Format: 300 trajectories for each target door in the training object set. The training object set consists of 42 cabinets. Each cabinet could contain multiple doors.
- Evaluation Protocol: This task has a held-out test object set (10 unseen cabinets). The evaluation protocol is similar to *OpenCabinetDrawer*.
- Task-specific Extra Observations: Since one cabinet can contain several doors, we specify the target door by a segmentation mask.

**PushChair**

- Objective: A dual-arm mobile robot needs to push a swivel chair to a target location on the ground (indicated by a red hemisphere) and prevent it from falling over. The friction and damping parameters for the chair joints are randomized.
- Success Metric: The chair is close enough (within 15 cm) to the target location, is static, and does not fall over.
- Demonstration Format: 300 trajectories for each chair in the training object set. The training object set consists of 26 chairs.
- Evaluation Protocol: This task has a held-out test object set (10 unseen chairs). The evaluation protocol is similar to *OpenCabinetDrawer*.
- Task-specific Extra Observations: None.

**MoveBucket**

- Objective: A dual-arm mobile robot needs to move a bucket with a ball inside and lift it onto a platform.
- Success Metric: The bucket is placed on or above the platform at the upright position, and the bucket is static, and the ball remains in the bucket.
- Demonstration Format: 300 trajectories for each bucket in the training object set. The training object set consists of 29 buckets.
- Evaluation Protocol: This task has a held-out test object set (10 unseen buckets). The evaluation protocol is similar to *OpenCabinetDrawer*.
- Task-specific Extra Observations: None.

# D  SOFT-BODY DETAILS

## D.1  SOFT-BODY SIMULATION AND 2-WAY COUPLING ALGORITHM

The simulation and 2-way coupling algorithm are summarized in algorithm 1.

---
**Algorithm 1** Rigid MPM Simulation and Dynamic Coupling

---
    initialize rigid scene
    initialize soft scene
    copy rigid body shapes and center of mass to soft scene
    initialize renderer
    **for** environment step **do**
        execute ManiSkill2 controllers
        **for** rigid step per environment step **do**
            process ManiSkill2 substep
            step rigid scene
            copy rigid body poses to soft scene
            initialize force-torque buffers per rigid body
            **for** soft step per rigid step **do**
                compute penalty forces
                accumulate equivalent rigid-body forces and torques in force-torque buffers
                MPM particle to grid (mass, momentum, forces)
                MPM grid compute velocity
                MPM grid to particle (velocity)
                integrate MPM particles
            **end for**
            apply accumulated forces and torques on rigid bodies
        **end for**
        copy rigid body states to SAPIEN renderer
        copy MPM particles to SAPIEN renderer
        execute renderer
    **end for**

---

## D.2  SOFT-BODY RENDERING

To support visual learning, we extended SAPIEN's renderer to support rendering particle-based soft body. To render particle-based material, one common approach is to convert the particles to triangle meshes using a meshing algorithm such as marching cubes; PlasticineLab implements a ray-tracing framework that renders spheres directly. Our approach is screen-space splatting (Cords & Staadt, 2008), similar to Nvidia Flex's built-in renderer. We customize SAPIEN's shader to render soft-body particles as spheres, use a bilateral filter to smooth the depth buffer, then compute normal and lighting on the smoothed soft-body depth. These are implemented as extra screen-space render passes. The effect of the smoothing filter is shown in figure 6. Moreover, we customize Warp to support the transfer of particle positions from simulation to renderer with a single GPU-GPU copy;

this further reduces rendering latency. A concern of the screen-space splatting is the inconsistency across different views due to the use of screen-space filters. However, in practice, by scaling the bilateral filter according to pixel distance from camera, the rendering results produced are visually consistent most of the time.

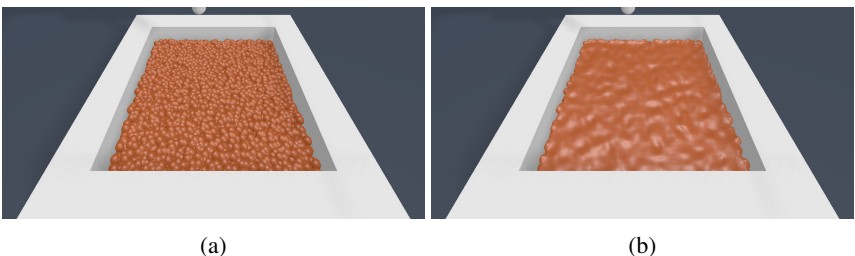

|              (a)              |              (b)              |

Figure 6: Our particle renderer (a) without bilateral filter, and (b) with bilateral filter.

### D.3   SOFT BODY DETAILS

We summarize the key parameters In table 4. For all soft body environments with rendering, a single environment runs at around 17-18 FPS; 16 parallel environments on a single GPU gives overall 80-84 FPS on a single RTX Titan GPU (4x real time). This performance gain is mainly due to the sequential CPU-rigid-body and GPU-soft-body simulation. It is also partially due to a single CPU processor not able to submit CUDA kernels fast enough to keep up with the GPU. Therefore, it can potentially be further optimized with vectorized simulation, which we leave as future work.

| Grid Length | Particle Volume | Density | Young's Modulus | Poisson Ratio | Yield Stress |
|---|---|---|---|---|---|
| 0.005 to 0.015 | 6.2e-8/1.2e-7 | 300 to 3000 | 1e4/3e5 | 0.3 | 2e3/1e4 |

Table 4: Ranges for key parameters used in our MPM simulation. All fields are in SI units.

## E   PERFORMANCE OPTIMIZATION DETAILS

### E.1   RENDER SERVER IMPLEMENTATION

Our render server is implemented with the gRPC framework, which exchanges Protocal Buffers with the HTTP 2 protocol over networks or unix sockets. The server side is managed by a thread pool, listening to client requests on multiple concurrent threads. For a "take picture" request from a worker process, our implementation puts the task of updating GPU matrices and launching draw calls onto another thread and returns immediately back to the worker process. This ensures minimum waiting time on the worker side.

On the rendering server, all rendering resources are managed by a central resource manager. Any resource loading request (e.g., models, images, textures) must go through the manager. The manager ensures only one copy of any resource is loaded onto the GPU, shared across potentially multiple scenes.

### E.2   ADDITIONAL BENEFITS OF RENDER SERVER

In general, rendering resource sharing across processes is not well supported. Rendering frameworks like OpenGL and Vulkan are designed to be efficient in a single-process multi-threaded environment. Resources and documentations on multi-process renderer are rarely provided. Therefore as a developer, it is far easier to understand renderers in a single process setting.

In addition, Nvidia's GPU profiler, Nsight System, is currently unable to profile Vulkan in multiple processes on Linux in our experiments. Running the renderer in multiple processes makes it hard to understand GPU performance. Thus, running rendering in one main process is almost required for a developer to optimize for hardware utilization.

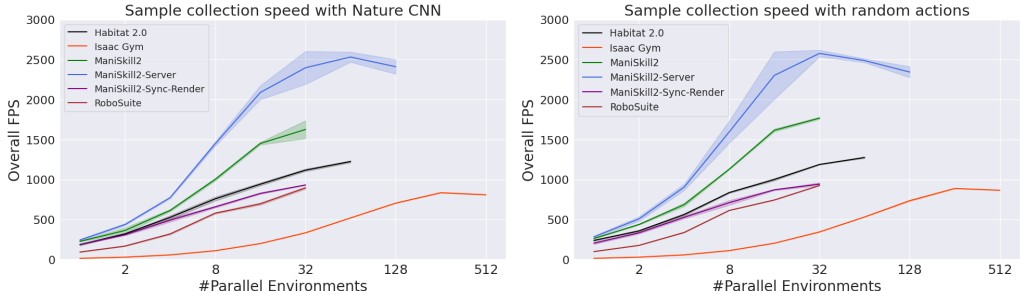

Figure 7: Comparison of sample collection speed (FPS) with random actions and with Nature CNN-sampled actions across different frameworks and different numbers of parallel environments. "ManiSkill2-Sync-Render" refers to ManiSkill2 with synchronous rendering and without render server. "ManiSkill2" refers to ManiSkill2 with asynchronous rendering but without render server. "ManiSkill2-Server" refers to ManiSkill2 with both asynchronous rendering and render server enabled. Some curves are not fully drawn beyond a certain number of parallel environments due to performance drop ("ManiSkill2-Server" & "Isaac Gym"), GPU out of memory ("RoboSuite" & "ManiSkill2-Sync-Render" & "ManiSkill2"), crash ("Habitat 2.0").

### E.3 MORE DETAILS ON SAMPLE COLLECTION SPEED COMPARISON

We use the *PickCube* environment to compare the sample collection speed between different simulators. The resulting total number of vertices of collision meshes is 1009 for ManiSkill2 and Habitat-2.0 and 1210 for IsaacGym and Robosuite. The number of vertices of visual meshes is 69747 for all simulators. Each control step consists of 25 physical simulation steps. We use the GPU pipeline for rendering. All frameworks are given a budget of 16 CPU cores(logical processors) of an Intel i9-9960X CPU with 128G memory and 1 RTX Titan GPU with 24G memory.

To complement results in Table 1, we plot further details about the relationship between the number of parallel environments and the policy sample collection speed across different frameworks in Figure 7. We adopt an agent that outputs random actions, along with a CNN-based agent that uses a randomly-initialized nature CNN (Mnih et al., 2015) as its visual backbone. We observe that ManiSkill2 with asynchronous rendering enabled (and without the render server) is already able to outperform the speed of other frameworks. With render server enabled, ManiSkill2 further achieves 2000+ FPS with 16 parallel environments on a single GPU.

## F ADDITIONAL EXPERIMENT DETAILS, RESULTS, AND ANALYSIS

### F.1 CONTACT-GRASPNET FOR *PickSingleYCB*

We directly use the pretrained model provided by the original authors of Contact-GraspNet (Sundermeyer et al., 2021). We exemplify successful trajectories by the model in Fig. 8, as well as common failure modes in Fig. 9.

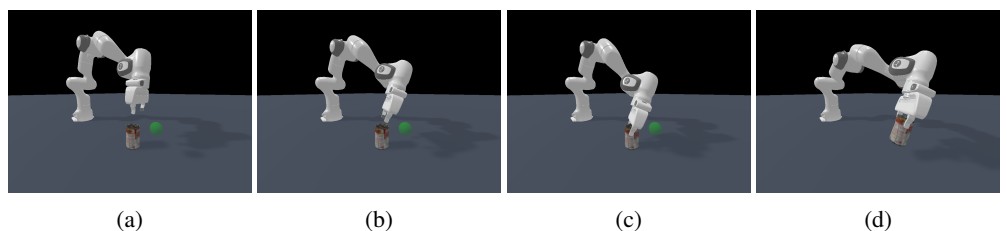

|       (a)       |       (b)       |       (c)       |       (d)       |

Figure 8: Sampled frames demonstrating a correct and successful grasp of a can. Frame (a) shows the initial state; (b) shows the gripper approaching the predicted grasp from above; (c) shows the gripper grasping the can; (d) shows the robot moving the can towards the green goal position.

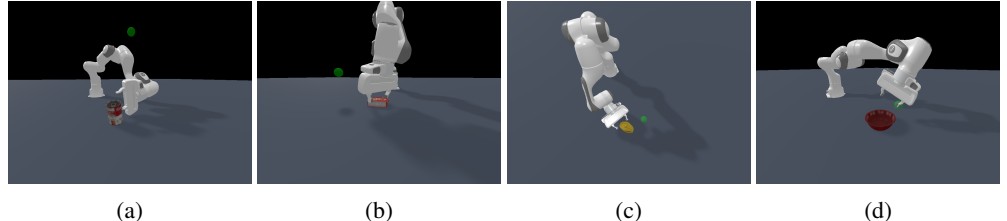

| (a) | (b) | (c) | (d) |

Figure 9: Examples of unsuccessful grasps. (a) shows an erroneous rotation prediction in grasp pose; (b) shows a correct rotation prediction in grasp pose, but the gripper is not close enough to the object to grasp it; (c) shows a reasonable grasp pose, but the gripper will slip away from the bottle upon finger closure due to friction and bottle geometry; (d) shows a reasonable grasp pose that is not achievable through motion planning due to kinematic constraints of the robot arm.

## F.2 TRANSPORTER NETWORK FOR *AssemblingKits*

| Gripper Type | Success (ours) | Success (Zeng et al.) | Pos < 5mm | Pos < 2.5mm | Pos < 1mm | Rot < 4° | Rot < 1° | Rot < 0.25° |
|---|---|---|---|---|---|---|---|---|
| Two-finger Gripper | 0.18 | 0.98 | 0.98 | 0.80 | 0.30 | 0.96 | 0.48 | 0.22 |
| Suction Gripper | 0.15 | 0.93 | 0.89 | 0.66 | 0.18 | 0.91 | 0.48 | 0.22 |

| Gripper Type | Success (ours) | Success (Zeng et al.) | Pos < 5mm | Pos < 2.5mm | Pos < 1mm | Rot < 4° | Rot < 1° | Rot < 0.25° |
|---|---|---|---|---|---|---|---|---|
| Two-finger Gripper | 0.18 | 0.99 | 0.99 | 0.82 | 0.31 | 0.96 | 0.48 | 0.24 |
| Suction Gripper | 0.14 | 0.96 | 0.92 | 0.66 | 0.17 | 0.94 | 0.52 | 0.31 |

Table 5: Success rate of Transporter Networks on our AssemblingKits task on training assets (up) and test assets (down). We also report the success rate based on the original paper, where a trial is successful if the target piece is placed within 1cm and 15 degrees of the goal position.

We adopt the official code from Zeng et al. (2020). To encourage precise rotation prediction, we increase the number of rotation prediction bins from 36 in the original work to 144. We further benchmark with two grippers: a suction gripper following the original work, and another two-finger gripper from Franka Panda. Our training dataset is generated from random initial configurations of training assets in AssemblingKits. More specifically, for each sampled initial configuration, we capture RGBD images from the hand-camera and the base-camera, unproject them to 3D point clouds, fuse the point clouds, and render the top-down orthographic image to feed to the Transporter Network model. The ground truth labels, namely the object pick position and the goal pose to place the object, are directly obtained from the environment state information.

Table 5 shows the results. The success rate of Transporter Network following our success criterion (which requires pieces to fully fit in holes) is a lot lower than following the metric in the original paper. We observe that failure modes are mainly due to imprecise rotation/position prediction for placing the target piece.

## F.3 DETAILED SETUP FOR IMITATION LEARNING & RL FROM DEMONSTRATIONS

In ManiSkill2, our demonstrations are provided using the *joint position* controller. Before we train demonstration-based agents on rigid & soft-body tasks, we first use the approach in Sec.2.2 to translate the provided demonstrations into the *delta end-effector pose* controller for rigid-body environments, and into the *delta joint position* controller for soft-body environments. We then filter the translated trajectories, such that only successful trajectories are used for agent learning.

For RGBD-based agents, we use IMPALA (Espeholt et al., 2018) as the visual backbone, and we concatenate images captured from the base camera and the hand camera as visual input. Image resolution is 128x128. For point cloud-based agents, we use PointNet (Qi et al., 2017) as the visual backbone. We first obtain a single fused point cloud by transforming point clouds from different cameras into the robot-base frame and concatenating the points together. We then remove the ground using height clip and randomly downsample the point cloud to 1200 points. For rigid-body environments, we also transform the point cloud (along with robot proprioceptive information

| Obs. Mode | PickSingleYCB | AssemblingKits | TurnFaucet | AvoidObstacles |
|---|---|---|---|---|
| Point Cloud | $0.36 \pm 0.06$ | $0.00 \pm 0.00$ | $0.01 \pm 0.01$ | $0.00 \pm 0.00$ |
| RGBD | $0.08 \pm 0.03$ | $0.00 \pm 0.00$ | $0.01 \pm 0.01$ | $0.00 \pm 0.00$ |

Table 6: Mean and standard deviation of success rates of DAPG+PPO on rigid-body tasks on held-out test objects. Training budget is 25M time steps.

and goal position, if given) into the end-effector frame. In addition, for environments where goal positions are given (PickCube and PickSingleYCB), we randomly sample 50 green points around the goal position to serve as visual cues and concatenate them to the input point cloud.

To train demonstration-based agents, for rigid-body environments, we use 1000 demonstration trajectories, except environments that have multiple assets (PickSingleYCB: 7300 trajectories; TurnFaucet: 4500 trajectories; AssemblingKits: 1700 trajectories). For soft-body environments, we use 200 demonstration trajectories (except *Pinch* with 1550 trajectories).

In this work, our demonstration-based online RL algorithm is modified from Proximal Policy Gradient (PPO) (Schulman et al., 2017) and Demonstration-Augmented Policy Gradient (DAPG) (Rajeswaran et al., 2017). We adopt the training objective modified from Jia et al. (2022). Here, we use $\rho_D$ and $\rho_\pi$ to denote the distribution of demonstration transitions and online environment rollout transitions, respectively. We can then write the overall policy loss as follows (value loss omitted here):

$$\mathcal{L}_\rho^{CLIP}(\theta) = -\mathbb{E}_{(s,a)\sim\rho} \left[ \min \left( \frac{\pi_\theta(a|s)}{\pi_{\theta_{old}}(a|s)} \hat{A}(s,a), \mathrm{clip}(r_t(\theta), 1-\epsilon, 1+\epsilon)\hat{A}(s,a) \right) \right]$$

$$\mathcal{L}_\rho^1(\theta) = -\mathbb{E}_{(s,a)\sim\rho}[\pi_\theta(a|s)]$$

$$\mathcal{L}_{DAPG+PPO}(\theta) = \mathcal{L}_{\rho_\pi}^{CLIP}(\theta) + \omega \cdot \mathcal{L}_{\rho_D}^1(\theta)$$

We set $\omega = 0.1 \cdot 0.995^N$, where $N$ is the epoch count for PPO. A PPO epoch is defined as online environment sampling steps followed by policy and value network updates.

For further details about networks and algorithm hyperparameters, see Appendix F.8.

### F.4   RESULTS FOR DAPG+PPO ON HELD-OUT OBJECT SETS

In Section 5.2, for tasks that have asset variations, we presented evaluation results on the training object set. In this section, we present results on the held-out object set. See Table 6 for details.

### F.5   FURTHER ANALYSIS OF IMITATION LEARNING ON SOFT-BODY TASKS

We observe that it is difficult for BC agents to accurately estimate the influence of an action on fine-grained soft body properties, such as displacement quantity and deformation. For example, both *Fill* and *Pour* tasks require a robot agent to move soft body objects (clay or liquid) into a target container, but *Fill* has a much higher success rate. An underlying cause is that *Fill* allows the robot agent to put all of the clay into the beaker while *Pour* requires higher precision i.e. the final liquid level must match the target line. Therefore, agents need to precisely control the bottle tilt angle in order to precisely control the amount of liquid poured into the beaker. Similarly, for *Excavate*, agents must perform fine judgments on how deep they should dig in order to scoop up a specified amount of clay. On the other hand, *Hang* does not require an agent to perform high accuracy measurements, so it is easier for agents to succeed.

In addition, we notice that BC agents cannot well utilize target shapes to guide precise soft body deformation. Specifically, for *Pinch* and *Write* that require shape deformation, the BC models have very poor performance. As shown in Fig. 10, the robot learns the motion of pinching and has made some progress toward the goal, but it is not good enough. Similarly, the robot agent has learned to draw some patterns, but they are not close enough to the target character.

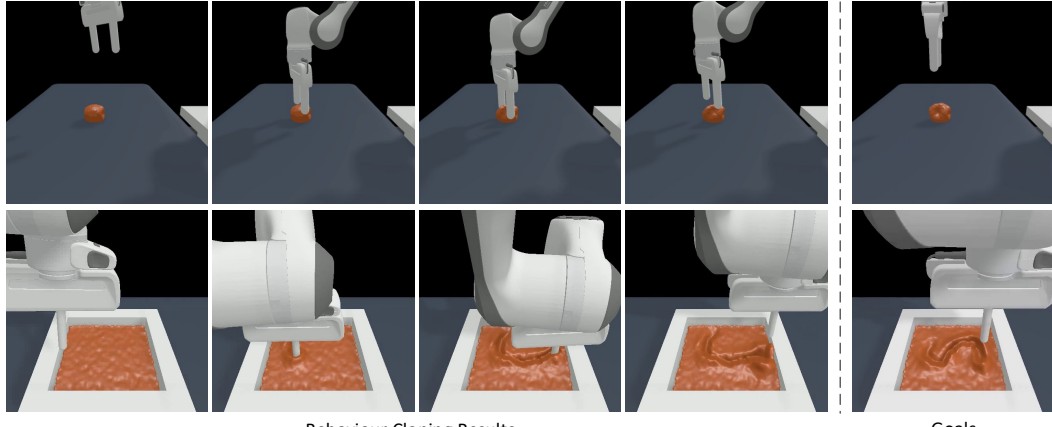

Behaviour Cloning Results                                                  Goals

Figure 10: Behaviour cloning examples for *Pinch* and *Write* tasks. BC models have learned to make some progress towards the goals but not enough to meet the success conditions.

| Original | *Delta Joint Position* Controller | Robot Base Frame Point Cloud | Remove Visual Goal Cues |
|---|---|---|---|
| $0.51 \pm 0.05$ | $0.22 \pm 0.18$ | $0.00 \pm 0.00$ | $0.16 \pm 0.07$ |

Table 7: Ablations on PickSingleYCB (training object set) for point cloud-based agents trained with DAPG+PPO. The "original" result refers to the result in Table 3, which uses the *delta end-effector pose* controller, transforms inputs point clouds into the robot's end-effector frame, and (for pick-and-place tasks where goal position is given) appends 50 green points sampled around the goal position to the input point clouds to serve as visual cues.

## F.6    MORE RESULTS ON POINT CLOUD-BASED MANIPULATION LEARNING

In this section, we examine factors that influence point cloud-based manipulation learning to complement the results in Section 5.2. Results are shown in Table 7. In addition to our prior observation that choices of controllers have a significant effect on performance, we also observe that (1) selecting good coordinate frames to represent input point clouds could be crucial for agents' success, which corroborates the findings in Liu et al. (2022); (2) adding proper visual cues could benefit point cloud-based agent learning.

## F.7    MORE ANALYSIS ON ASSEMBLY TASKS

In Section 5.2, we observed that agents trained with Imitation Learning or Reinforcement Learning perform poorly on many tasks that require high precision, such as the assembly tasks. We further analyze sources of difficulty from these tasks by training agents on easier versions of these tasks where the clearance threshold is significantly increased. Results are shown in Table 8. We observe that when we increase clearance threshold and decrease task difficulty, some agents are able to achieve a lot higher performance. However, even in this case, many agents still do not perform well. We conjecture that this is due to many other challenging factors, such as occlusions of the target slots when agents attempt to insert a peg or a charger.

## F.8    NETWORK ARCHITECTURES AND HYPERPARAMETERS FOR IL & RL

In this section, we present the specific network architectures and algorithm hyperparameters used for Section 5.2. Here we define "state vector" as the concatenation of proprioceptive robot state information and task-specific goal information (if given).

For IMPALA (Espeholt et al., 2018), the visual backbone is similar to ResNet10 (He et al., 2016), with hidden size in all layers equal to 64, normalizations removed, and the first convolution layer modified to have kernel size 4 and stride 4. Features from the last layer are projected to a 384-

| Observation Mode | PegInsertionSide(1x) | PegInsertionSide(10x) | PlugCharger(1x) | PlugCharger(10x) |
|---|---|---|---|---|
| Point Cloud | $0.00 \pm 0.00$ | $0.01 \pm 0.01$ | $0.00 \pm 0.00$ | $0.01 \pm 0.01$ |
| RGBD | $0.00 \pm 0.00$ | $0.00 \pm 0.00$ | $0.00 \pm 0.00$ | $0.00 \pm 0.00$ |

| Observation Mode | PegInsertionSide(1x) | PegInsertionSide(10x) | PlugCharger(1x) | PlugCharger(10x) |
|---|---|---|---|---|
| Point Cloud | $0.01 \pm 0.02$ | $0.74 \pm 0.10$ | $0.01 \pm 0.01$ | $0.02 \pm 0.02$ |
| RGBD | $0.01 \pm 0.01$ | $0.05 \pm 0.03$ | $0.01 \pm 0.01$ | $0.29 \pm 0.07$ |

Table 8: Analysis of IL & demonstration-based RL on assembling tasks. We control the difficulty of tasks by changing clearance configurations. Top: Behavior Cloning. Bottom: DAPG+PPO. 1x=default clearance (3mm for PegInsertionSide and 0.5mm for PlugCharger); 10x = 10 times default clearance.

dimensional vector before being concatenated with the state vector. For PointNet (Qi et al., 2017), the hidden layer sizes are [64, 128, 512] before maxpooling over the number-of-points dimension. We do not use spatial normalization layers, and we add layer normalization to the output of each intermediate layer before maxpooling.

For both BC and DAPG+PPO, we use learning rate of 3e-4 for training and optimize with Adam (Kingma & Ba, 2015). For DAPG+PPO, the actor and critic networks share the same visual backbone. In addition, when training point cloud-based policies, we initialize the linear layer right before the policy and value head output to be zero. We find this helpful for stabilizing point cloud-based policy learning. Hyperparameters are shown in Table 9.

| Hyperparameters | Value |
|---|---|
| Discount ($\gamma$) | 0.95 |
| $\lambda$ in GAE | 0.95 |
| PPO clip range | 0.2 |
| Max KL | 0.1 |
| Gradient norm clipping | 0.5 |
| Entropy | 0.0 |
| Number of samples per PPO step | 20000 |
| Number of samples per minibatch | 300 |
| Number of critic warmup epochs | 4 |
| Number of PPO update epochs | 2 |
| Critic loss coefficient | 0.5 |
| Demonstration loss coefficient | $0.1 \cdot 0.995^N$ |
| Recompute advantage after each PPO update epoch | True |
| Reward normalization | True |
| Advantage normalization | True |
| Only use critic loss to update visual backbone | True |
| Reset environment upon success during policy rollout | False |

Table 9: Hyperparameters for DAPG+PPO.

# G    COMPARISON WITH OTHER BENCHMARKS FOR ROBOTIC MANIPULATION

| Benchmark | **ManiSkill2** | BEHAVIOR-1K[3] | Habitat 2.0[4] | IsaacGym[5] | ManipulaThor[6] | MetaWorld | Robosuite | RLBench | TDW[7] |
|---|---|---|---|---|---|---|---|---|---|
| Grasp implementation | Physical | Abstract | Abstract | Physical | Abstract | Physical | Physical | Abstract | Abstract |
| #Demo trajectories | **>30k** | - | - | - | - | Procedural | ∼2000 | Procedural | - |
| Multi-controller support | **Yes** | Unknown | **Yes** | No | No | No | **Yes** | **Yes** | No |
| Visual RL/IL baselines | **Full** | Limited | **Full** | No | **Full** | No | No | No | No |
| #Object models | >2144*[8] | 3324 | YCB | - | 150 | 80 | 10 | 28 | 112 |
| #Scenes | - | **306** | 105 | - | 30 | - | - | - | 105 |
| Ray-tracing support | Kuafu | Omniverse | - | - | - | - | NVISII | - | Unity |
| Domain randomization | Partial[9] | Unknown | No | **Yes** | No | No | **Yes** | **Yes** | No |
| Rigid-body simulation | SAPIEN | Omniverse | Bullet | PhysX 5 | Unity | Mujoco | Mujoco | V-REP | Unity |
| Soft-body simulation | Warp-MPM[10] | Omniverse | - | - | - | - | - | - | - |

Table 10: Comparison with other existing benchmarks for robotic manipulation. The information of each benchmark is based on its major focus. For example, all the simulation backends of these benchmarks can support physically implemented grasp, but some of them focus on high-level actions and thus use abstract grasp for benchmarking algorithms. *Multi-controller support* measures whether a benchmark has implemented multiple controllers and provided interfaces to select one for each task.

Table 10 compares ManiSkill2 with other existing benchmarks for robotic manipulation. ManiSkill2 features large-scale demonstrations for every task, a great variety of objects, multi-controller support and conversion of demonstration action spaces, and a focus on fully physically implemented grasp. We invested significant efforts to select, fix, and re-model objects and integrate them to our task families, generate demonstrations with fully physical grasping, and perform large-scale visual manipulation benchmarking. Through these processes, we carefully verify all of our tasks. The low success rates on many tasks demonstrate that our benchmark poses interesting and challenging problems for the community.

We are actively working on current limitations of ManiSkill2: photorealism, domain randomization, and scene-level variation. First, the simulation backend (SAPIEN) of ManiSkill2 supports a ray-tracing renderer (Kuafu). However, photorealism is usally achieved at the cost of speed. For example, BEHAVIOR-1K Li et al. (2022) can only achieve 60 FPS with Nvidia Omniverse and require high-end GPUs. In this work, we focus on physically realistic short-horizon and low-level visuomotor manipulation. We have experimentally supported ray-tracing rendering for use cases like generating data offline for supervised learning or fine-tuning a policy learned on non-photorealistic data. Second, we have included domain randomization for physical parameters and visual appearance for some of tasks. We will provide tutorials for users to customize domain randomization according to their use cases. Third, we are introducing mobile manipulation tasks similar to those in ManipularTHOR ArmPointNav (Ehsani et al., 2021) and Habitat 2.0 (Szot et al., 2021) but with physically implemented base movement and grasp. Those tasks will involve scene-level variations and demand advanced navigation abilities.

---

[3]Since BEHAVIOR-1K (Li et al., 2022) is not public yet, some detailed information is unknown.

[4]For Habitat 2.0 (Szot et al., 2021), we refer to its Home Assistant Benchmark.

[5]For IsaacGym (Makoviychuk et al., 2021), we refer to its 4 manipulation tasks in the original paper: Franka Cube Stacking, Shadow Hand, Allegro Hand, and TriFinger.

[6]For ManipularTHOR (Ehsani et al., 2021), we refer to its ArmPointNav task.

[7]For TDW (Gan et al., 2020), we refer to its Transport Challenge (Gan et al., 2021)

[8]Assets like boards with slots for *AssemblingKits*, feasible layouts of obstacles for *AvoidObstacles*, and target characters for *Write* are not included. All of these assets require offline generation.

[9]ManiSkill2 has included some domain randomization for physical parameters (e.g., joint damping and friction of *PushChair*, *OpenCabinetDoor*, and *OpenCabinetDrawer*) and visual appearance (e.g., colors in *PickSingleEGAD* and *PandaAvoidObstacles*).

[10]ManiSkill2 implements a MPM-based soft-body simulator based on Warp (Macklin, 2022). Deformable objects like cloth are not supported yet.

## H    CONTRIBUTIONS

- Designed and implemented ManiSkill2 infrastructure: Jiayuan Gu, Fanbo Xiang, Rui Chen
- Designed rigid-body environments and collected demonstrations: Jiayuan Gu, Xuanlin Li, Xiqiang Liu, Tongzhou Mu
- Designed soft-body environments and collected demonstrations: Fanbo Xiang, Xinyue Wei
- Designed and implemented server-based asynchronous rendering: Fanbo Xiang, Zhan Ling, Jiayuan Gu
- Designed and implemented the soft-body simulator: Fanbo Xiang, Xinyue Wei, Xiaodi Yuan
- Conducted sense-plan-act experiments: Stone Tao, Xiqiang Liu, Jiayuan Gu
- Conducted RL/IL experiments on rigid-body environments: Yunchao Yao, Xuanlin Li, Zhan Ling
- Conducted IL experiments on soft-body environments: Yihe Tang, Xinyue Wei, Zhan Ling
- Conducted real-world experiments: Rui Chen, Pengwei Xie
- Implemented the cloud-based evaluation system: Stone Tao, Fanbo Xiang
- Managed or advised on the project: Hao Su, Rui Chen, Zhiao Huang

