# OpenReview forum: "ManiSkill2: A Unified Benchmark for Generalizable Manipulation Skills"
_ICLR.cc/2023/Conference — ICLR 2023 poster_

### Official Review · Reviewer_zjV2 · 2022-10-13

**Confidence:** 4
**Correctness:** 2
**Technical Novelty And Significance:** 3
**Empirical Novelty And Significance:** 3
**Recommendation:** 6

**Clarity, Quality, Novelty And Reproducibility:**

**R1.** On reproducibility: I appreciate the release of all the code and simulator functionality. A lot of work has gone into this, and it will benefit the community.

**R2.** Much of the work seemed to focus on clear next steps from ManiSkill 1.0. The authors make many incremental improvements, which in aggregate will be helpful to the community.

**Strength And Weaknesses:**

### Strengths
* The extent of the manipulation capabilities and support is great.
* The number of objects used for a manipulation benchmark is quite impressive.
* The performance benchmarks and simulation speeds are impressive. I appreciate all the engineering effort that went into this paper!
* The sim2real results are neat.


### Weaknesses

* **W1. Lack of discussion on objects.** The authors claim they have 2000+ object models, a significant increase over the 162 objects in ManiSkill 1.0. However, there isn't much of a discussion on where these objects come from. Where do they come from? Are they going to be released (and under what license)? What is the distribution of the object categories? To what extent are they interactive/articulated?

* **W2.** The backgrounds and setup of the scenes are quite weak. In comparison to ManipulaTHOR, Habitat 2.0, and iGibson 2.0 for manipulating different objects, which are in household environments, the gray backdrop behind the objects makes the environments feel like a toy environment. It would be much better if the manipulation of objects happened in the context of other objects (e.g., when turning on a faucet, the agent is always in a bathroom/kitchen), when opening the cabinet, the agent is always in a kitchen or bedroom, etc. It doesn't seem like the tasks are very realistic to what one might want a robot to do in the real-world. The one task used for sim2real in this paper is merely picking up a fairly visually distinct cube.

* **W3.** I wish the paper was formatted a bit differently, and instead focused mostly on experiments in the main part of the paper, instead of discussing technical details in sections 3 and 4. The main part of the paper doesn't seem like much of the content in these sections is necessary for most users of ManiSkill2 to read. It seems like much of this could have gone into the appendix.

**Summary Of The Paper:**

The authors propose ManiSkill2, a benchmark for robotic manipulation. It builds upon ManiSkill 1.0 and has better support for different types of gripper-based manipulations, uses significantly more interactive objects, runs much faster, and includes soft-body simulations. The authors show experiments on a few tasks, 4+ million frames of gripper-based interactions, which can be used for imitation learning, and include pickup object sim2real results.

**Summary Of The Review:**

The engineering effort that went into this is quite impressive to support such a wide variety of agent-based manipulations. However, the work is pretty incremental, focusing on a few clear next steps from the previous work in ManiSkill 1.0. I would have also liked the authors to start moving towards more realistic task setups that a robotic agent might actually encounter in the real-world. Currently, the agents are just operating with isolated objects on a grayscale background, and this isn't practically useful for most tasks we want robots to perform.

---

> ### Author Response · Authors · 2022-11-16
> **Response**
>
> We sincerely thank you for your constructive feedback! We address the comments and questions below.
>
>
> > Lack of discussion on objects. Where do they come from? Are they going to be released (and under what license)? What is the distribution of the object categories? To what extent are they interactive/articulated?
>
> Thank you for the comment. We refer the reviewer to Appendix C in the revision for detailed descriptions of assets used in each task. In brief, our objects consist of 74 YCB objects and 40 held-out rigid objects(`PickSingleYCB`, `PickClutterYCB`), 1750 EGAD objects (`PickSingleEGAD`, `Pinch`), 78 PartNet-mobility articulated faucets (`TurnFaucet`), in addition to 162 PartNet-mobility articulated objects from ManiSkill1. Besides, we provide over 300 kit configurations (generated from 20 training and 20 testing shapes) for `AssemblingKits`, over 1700 feasible layouts of obstacles for `AvoidObstacles`, over 1500 target shapes for `Pinch`, and over 50 characters for `Write`. All the assets are interactive and will be released under CC BY-NC 4.0 unless otherwise specified.
>
> > The backgrounds and setup of the scenes are quite weak.
>
> Thank you for the suggestion. We will provide tutorials to add static non-interactive background scenes (e.g., kitchen, bathroom, bedroom) for manipulation tasks. However, we would like to address that learning generalizable manipulation policies for diverse objects **without abstract grasp** is a very challenging problem itself. Existing benchmarks like ManipulaTHOR, Habitat 2.0, and iGibson 2.0, use abstract grasp. BEHAVIOR-1K "observes a radical effect of removing the simplifying assumptions for grasping, leading to failures in all activities due to missed grasps." Habitat 2.0 "finds the success rate without abstracted grasping is 20.16% versus 89.70% with abstracted grasping." Thus, existing benchmarks instead focus on long-horizon interactive navigation with diverse visual apperance. On the other hand, we deliberately avoid introducing more complexity in ManiSkill2, to faciliate the community to diagnose key challenges for manipulation. For example, through our experiments on Assembly environments (*PegInsertionSide*, *PlugCharger*, and *AssemblingKits*), we observe that current methods are insufficient for performing highly-precise controls. We also plan to add long-horizon mobile manipulation with physically implemented base movement and grasp in the future.
>
>
>
> > I wish the paper was formatted a bit differently, and instead focused mostly on experiments in the main part of the paper, instead of discussing technical details in sections 3 and 4.
>
>
> The main results from our experiments (Section 5 and Appendix F) are as follows:
>
> - Prior works (Transporter Network, point cloud / RGBD-based imitation learning & online RL) struggle at our assembling tasks (e.g., *PegInsertionSide*, *PlugCharger*, *AssemblingKits*). This suggests that existing RL algorithms might have been insufficient yet to perform highly precise controls, and our benchmark poses meaningful challenges for the community.
> - Using point clouds as visual representation could lead to better performance on pick-and-place tasks (e.g., PickCube, StackCube, PickSingleYCB) than using RGB-D images as visual representation.
> - For soft-body tasks, it is challenging for agents to accurately estimate the influence of an action on fine-grained soft-body properties, such as quantity and deformation, resulting in low performance on tasks such as *Pour* and *Excavate*. In addition, it is challenging to use target shapes to guide precise soft-body deformation (the success rates are zero for *Pinch* and *Write* even though the agent has made some progress towards a goal)
>
> We'd be happy to move up some results in the appendix to the main part of the paper.

---

> > ### Comment · Reviewer_zjV2 · 2022-11-19
> > **Response to authors**
> >
> > Thank you for the response in clarifying objects, discussing backgrounds, and formatting. I think moving some of the experiments up to the main part of the paper would be good for the camera ready. Nevertheless, I'm impressed by the work and will increase my rating accordingly.

---

### Official Review · Reviewer_fwz3 · 2022-10-23

**Confidence:** 4
**Correctness:** 3
**Technical Novelty And Significance:** 3
**Empirical Novelty And Significance:** 4
**Recommendation:** 8

**Clarity, Quality, Novelty And Reproducibility:**

This paper is sufficiently novel and clearly written, with abundant technical details presented.

**Strength And Weaknesses:**

Strengths:
- Variety of the manipulation tasks: stationary/mobile-base, single/dual-arm, rigid/soft-body, different action spaces and controllers. These, in combination, provide richer families of tasks than previous benchmarks.
- Good software engineering techniques for efficient learning with multi-physics and visual input.

Weaknesses:
- Lack of different grippers and arms support - only Panda arm and two-finger/suction grippers are implemented. To facilitate dexterous manipulation and more sim-to-real transfer, it is important to support more industrial arms (IIWA, UR5e, etc.) with multi-finger grippers.
- Limitations are not presented in the paper. It would be important to understand what are the pros and cons of this benchmark compared to existing ones in a clear manner (e.g. having a table showing the comparison with popular benchmarks on multiple dimensions/features). For example, Maniskill2 does not seem to support photorealism, domain randomization, cloth simulation, scaling to thousands of parallel environments (like Isaac Gym), etc., which exist in some of the previous benchmarks.

Questions:
- While claiming Maniskill2 is the first to support real-time simulation and rendering of MPM material, I don't see concrete statistics supporting this. What is the speed comparison against naive MPM simulation or PlasticineLab?
- Is two-way rigid-soft coupling necessary for manipulation tasks in Maniskill2? Though I believe it is more realistic (compared to one-way coupling), for manipulation purposes, the rigid grippers are usually much heavier than the soft materials being manipulated. In this case, maybe one-way coupling is realistic enough and more efficient. It would be necessary to showcase where the two-way coupling is a must for some manipulation tasks.

**Summary Of The Paper:**

This paper proposes a new benchmark for robotic manipulation that supports various types of manipulation tasks with improved efficiency. The main contributions are:
- Rich task collections, assets, and demonstrations.
- Multi-controller support and action space conversion.
- Efficient 2-way coupling rigid-MPM simulation.
- Efficient parallelization for visual RL training.

**Summary Of The Review:**

I recommend acceptance based on the benchmark novelty and the engineering efforts for simulation efficiency. In my opinion, this paper has no major flaws, and I believe there is enough motivation for the community to use this benchmark to facilitate future research.

---

> ### Author Response · Authors · 2022-11-16
> **Response [2/2]**
>
> > Speed comparison against naive MPM simulation or PlasticineLab
>
> Naive MPM simulation: The naive MPM implementation, i.e., the original MLS-MPM implementation in C++, was deprecated in favor of their new Taichi implementation, which has been ported into PlasticineLab. Therefore, PlasticineLab is already an implementation with the best performance.
>
> PlasticineLab: at an early stage of ManiSkill2 development, we have integrated PlasticineLab with 2-way coupling, which runs at roughly 5 FPS per environment. After switching to our own Warp-based implementation with optimizations for data transfers, it runs at 19 FPS per environment, which is almost a 4x speed-up. In addition, we also significantly reduced the environment compilation and startup time of PlasticineLab (from >20s to <1s), which is crucial for development and debugging.
>
>
>
> > Is two-way rigid-soft coupling necessary for manipulation tasks in Maniskill2? Maybe one-way coupling is realistic enough and more efficient.
>
>
> One-way coupling requires many hand-crafted conditions to avoid instability. For example, in PlasticineLab, rolling pin is coded to have a smaller z-direction action than xy-direction actions because large z-direction motion without 2-way coupling would let the pin touch the ground without any resistance, completely flattening the soft body and crashing the simulation along the way. Such design is not scalable, as it requires environment developers to use heuristics to avoid crashing the simulator. With 2-way coupling, it requires many fewer task-specific heuristics to design an environment.
>
> In addition, the major overhead that can be avoided by one-way coupling is copying the force feedback from the soft-body simulation to the rigid-body simulation, which is not even the bottleneck of soft-body simulation. Moreover, such feedback (especially contact feedback) could be very helpful for reward design, demonstration generation, or environment debugging. Thus, fetching the force feedback is almost inevitable for a general design.

---

> ### Author Response · Authors · 2022-11-16
> **Response [1/2]**
>
> We sincerely thank you for your constructive feedback! We address the comments and questions below.
>
>
> > Lack of different grippers and arms support
>
> Thank you for the suggestion, and we plan to support dexterous hands in the next generation of ManiSkill2. Two-finger grippers are still the most popular and affordable hardware. Also, many research problems are unsolved -- when two-finger grippers are used in the scenarios of two-arm collaboration, 3D sensor input, soft-body manipulation, or manipulation of unseen objects, not only many challenges of policy learning  are still far from being solved, but even many issues of benchmark creation are not fully addressed (e.g., simulation parameter tuning, task reward design, and demo collection). Therefore, we choose to prioritize in building a solid benchmark for two-finger grippers in this work, and leave the support of dexterous hands to the future (current plan is the next year).
>
> To be more specific, the current ManiSkill2 benchmark focuses on the Panda arm due to its broad usage and high precision in both the simulation and real world. We have also included different end-effectors for soft-body tasks (e.g., bucket for *Fill* and *Excavate*; stick for *Write* and *Pinch*). Besides, we are actively working on supporting more arms and end-effectors, e.g., ROKAE xMate3 arm and Robotiq gripper used in our sim-to-real experiments. We will also provide detailed tutorials for users to add custom actuators.
>
> > Pros and cons (limitations) of this benchmark compared to existing ones. For example, Maniskill2 does not seem to support photorealism, domain randomization, cloth simulation, scaling to thousands of parallel environments (like Isaac Gym), etc., which exist in some of the previous benchmarks.
>
> Thank you for the suggestion. Please refer to Appendix G in the revision for the discussion on limitations and comparsion with existing benchmarks.
>
> We would like to briefly discuss some features mentioned by the reviewer.
> - Photorealism: The simulation backend (SAPIEN) of ManiSkill2 supports a ray-tracing renderer (Kuafu). However, photorealism is usally achieved at the cost of speed. For example, BEHAVIOR-1K can only achieve 60 FPS with Nvidia Omniverse and require high-end GPUs. In this work, we focus on physically realistic low-level visuomotor manipulation. We experimentally support ray-tracing rendering for use cases like generating data offline for supervised learning or fine-tuning a policy learned on non-photorealistic data.
> - Domain randomization: We have included some domain randomization for physical parameters (e.g., joint damping and friction of `PushChair`, `OpenCabinetDoor`, and `OpenCabinetDrawer`) and visual appearance (e.g., colors in `PickSingleEGAD` and `PandaAvoidObstacles`). Please see updated Appendix C for more task-specific details. We will also provide tutorials for users to customize domain randomization according to their use cases.
> - Cloth simulation: Our MPM-based soft-body simulator is not yet able to simulate cloth. We are integrating the recently open-sourced PhysX5 and working on a custom FEM-based simulator. We plan to support cloth simulation in the next version of ManiSkill2.
> - Scaling to thousands of parallel environments (like Isaac Gym): For state-based environments, Isaac Gym uses GPU-based simulation, and thus can achieve higher FPS with more parallel environments. For example, with 16384 parallel environments, the FPS can be 35744±30. However, for visual Isaac Gym environments, we found that scaling to over 512 parallel environments will harm the FPS. At 512 parallel environments, the visual environment FPS is 865±35, while the state environment FPS is 5807±111. This performance gap shows that the rendering process of visual observation generation is a great bottleneck on the speed of visual environments, which demonstrates the significance of our render-server in Section 4 of our main paper.

---

### Official Review · Reviewer_RgH8 · 2022-10-24

**Confidence:** 4
**Correctness:** 3
**Technical Novelty And Significance:** 2
**Empirical Novelty And Significance:** 2
**Recommendation:** 6

**Clarity, Quality, Novelty And Reproducibility:**

The paper is of good quality. The illustrations and experimental results are clear to me.

**Strength And Weaknesses:**

Strength:
1.	The proposed environment supports a wide range of observation and controller types and supports mainstream algorithms in sense-plan-act, RL and IL frameworks.
2.	The tasks include those from previous works and some newly proposed ones, covering a wide aspect of challenges in manipulation. The support for soft-body tasks is of great importance.
3.	The asynchronous rendering approach and render server implementation allow improved rendering performance and reduced memory usage.

Weaknesses:
1.	All the proposed manipulation tasks are based on robotic arms and lack support for other actuators (e.g. Dexterous hands).
2.	Since different simulators have different parameters that may drastically affect the overall performance, please include the details of parameters for each simulator (e.g. Did you use the GPU pipeline in IsaacGym or just used the CPU pipeline, the substeps for physics simulation, the number of the total vertex in the simulation).
3.	The performance comparison is carried out on RGBD input settings. The performance of pure state input and point-cloud input is not reported.
4.	The effectiveness and accuracy of demonstration conversion are neither discussed in detail nor measured quantitatively.
5.	Point-cloud observations in some tasks contain ground truth segmentation, which cannot be easily obtained in the real world. This barrier for sim-to-real transfer could be avoided with a better observation design.


**Summary Of The Paper:**

The paper presents a new benchmark environment for manipulation tasks. The environment supports both traditional rigid-body manipulation tasks and soft-body tasks, while unified interface allows a wide range of algorithms. Results show that the asynchronous rendering and render server approach provides an improved FPS for CNN-based policy learning.

**Summary Of The Review:**

This work proposes several improvements to the current simulator for manipulation. The improvements in rigid-body soft-body interface and simulation performance, along with the variety of supported tasks, is of great value. This work's weakness includes omitted details from different sections and an issue in observation. Overall, I am leaning towards acceptance.

---

> ### Author Response · Authors · 2022-11-16
> **Response [2/2]**
>
> > Effectiveness and accuracy of demonstration conversion are neither discussed in detail nor measured quantitatively.
>
> In the following table, we exemplify the success rate of our demonstration conversion method (Section 2.2) by converting from the *joint position* controller to the *delta end-effector pose* controller on 3 representative tasks. A demonstration is converted successfully if, following the same trajectory initialization and the converted actions, the task is successful at the last time step. We observe that our closed-loop demonstration conversion process achieves high success rates. Please refer to Appendix B.5 in the revision for more details.
>
>
> | PickSingleYCB | AssemblingKits | Write |
> | -------- | -------- | -------- |
> | 99%     | 98%     | 100%     |
>
>
> > Point-cloud observations in some tasks contain ground truth segmentation
>
> Different from the previous version, ManiSkill2 does **not** include ground-truth segmentation in the default observation modes (`rgbd` or `pointcloud`). All of our visual-based experiments do not leverage such privileged information. For example, to specify which link of a faucet should be manipulated in `TurnFaucet`, we use its initial position instead of a per-step ground-truth segmentation mask. Besides, we also support observations modes (`rgbd_robot_seg`, `pointcloud_robot_seg`) to provide the segmentation masks of robot links, which facilitates robotic applications and can be obtained in the real world using the robot proprioceptive information. We have clarified this in Appendix C.1 in the revision.

---

> ### Author Response · Authors · 2022-11-16
> **Response [1/2]**
>
> We sincerely thank you for your constructive feedback! We address the comments and questions below.
>
> > Lack support for other actuators (e.g. Dexterous hands)
>
> Thank you for the suggestion, and we plan to support dexterous hands in the next version of ManiSkill2. Two-finger grippers are still the most popular and affordable hardware. Also, many research problems are unsolved -- when two-finger grippers are used in the scenarios of two-arm collaboration, 3D sensor input, soft-body manipulation, or manipulation of unseen objects, not only many challenges of policy learning  are still far from being solved, but even many issues of benchmark creation are not fully addressed (e.g., simulation parameter tuning, task reward design, and demo collection). Therefore, we choose to prioritize in building a solid benchmark for two-finger grippers in this work, and leave the support of dexterous hands to the future (current plan is the next year).
>
> To be more specific, the current ManiSkill2 benchmark focuses on the Panda arm due to its broad usage and high precision in both the simulation and real world. We have included a custom mobile manipulator with Panda arms for 4 tasks migrated from the previous generation of ManiSkill. We have also included different end-effectors for soft-body tasks (e.g., bucket for *Fill* and *Excavate*; stick for *Write* and *Pinch*). Besides, we will support more arms and end-effectors in our public release, e.g., ROKAE xMate3 arm and Robotiq gripper used in our sim-to-real experiments. We will also provide detailed tutorials for users to add custom actuators.
>
>
>
> > Details of parameters for each simulator being compared (e.g. Did you use the GPU pipeline in IsaacGym or just used the CPU pipeline, the substeps for physics simulation, the number of the total vertex in the simulation)
>
> We have added further details in Appendix E.3 in the revision. We used the GPU pipeline for rendering in all the simulation frameworks. Each control step consists of 25 physical simulation steps. The number of vertices of collision meshes (robot and objects) is 1009 for ManiSkill2 and Habitat-2.0 and 1210 for IsaacGym and Robosuite. The number of vertices of visual meshes is 69747 for all simulators.
>
> > Performance of pure state input and point-cloud input is not reported
>
> To benchmark trajectory sampling speed with point cloud observations, we adopt the same camera setting (128x128 resolution) as with image observations.  We use the same computation budget as that for image-based benchmarking (16 CPU cores (logical processors) of an Intel i9-9960X CPU with 128G memory and 1 RTX Titan GPU with 24G memory). After rendering an RGB-D image, we backproject it into a point cloud. We then post-process the point cloud following the same setting as our experiments in Section 5.2. Specifically, we first remove the ground (i.e., remove the points whose z-coordinate < 1e-3), and then randomly downsample the resulting point cloud to 1200 points. Using our render server, the FPS is 2298±34 with random actions, and 2237±106 with PointNet.
>
> In addition, we benchmark the speed of state-based environments using the same computation budget as before. The FPS of ManiSkill2 with random actions and with 64 parallel environments is 2946±147. The FPS of RoboSuite with 32 parallel environments is 991±8. The FPS of Habitat with 64 parallel environments is 3500±62. For Isaac Gym with 512 parallel environments, the FPS is 5807±111. However, as previously shown in Table 1 of our paper, when we create ***visual*** Isaac Gym environments instead of state-based Isaac Gym environments, the FPS drops to 865±35. This performance gap shows that the rendering process of visual observation generation is a great bottleneck on the speed of visual environments, which demonstrates the significance of our render-server in Section 4 of our main paper.
>
> As a side note, Isaac Gym uses GPU-based simulation for state-based environments, and thus can achieve even higher FPS with more parallel environments. For example, with 16384 parallel environments, the FPS can be 35744±30. However, for visual Isaac Gym environments, we found that scaling to over 512 parallel environments will harm the FPS. In addition, we cannot create as many parallel visual environments as state-based environments. In fact, 3072 parallel visual Isaac Gym environments will cause out-of-memory errors on GPU. Similar observations are also reported by [Bi-DexHands](https://github.com/PKU-MARL/DexterousHands/blob/main/docs/point-cloud.md).

---

### Decision · Program_Chairs · 2023-01-20

**Decision:**

Accept: poster

**Justification For Why Not Higher Score:**

Scope of contribution.

**Justification For Why Not Lower Score:**

Core quality and relevance for this audience.

**Metareview: Summary, Strengths And Weaknesses:**

The authors identify a core part of embodied AI to be manipulation of diverse objects and generalization of manipulation abilities across object instances.  They introduce a new suite of manipulation tasks with object-level variations to support research on these challenges.  The aim of this task suite is to support vision-based policies and fast simulation.  This is done using Warp, a JIT framework from Nvidia that allows for specification of tasks using python, but faster execution.  A speed comparison is performed, demonstrating good performance relative to relevant alternatives.  Demonstrations are also included, and tests demonstrating the plausibility of sim2real are also included in the paper.

The reviewers all found this paper and benchmark to be a valuable contribution to the field.  Though there were some issues raised about performance comparisons details, features (e.g., gripper options and visual backgrounds), omitted details (e.g. objects included), etc, these were reasonably well addressed by author responses.

**Note From Pc:**

if the above contains the word "oral" or "spotlight" please see: "oral" presentation means -> notable-top-5% and "spotlight" means -> notable-top-25%. As stated in our emails, we are disassociating presentation type from AC recommendations